# Randomized Truthful Auctions with Learning Agents

**Gagan Aggarwal**
Google Research
gagana@google.com

**Anupam Gupta**
New York University, Google Research
anupam.g@nyu.edu

**Andres Perlroth**
Google Research
perlroth@google.com

**Grigoris Velegkas**[*]
Yale University
grigoris.velegkas@yale.edu

## Abstract

We study a setting where agents use no-regret learning algorithms to participate in repeated auctions. Kolumbus and Nisan (2022a) showed, rather surprisingly, that when bidders participate in second-price auctions using no-regret bidding algorithms, no matter how large the number of interactions $T$ is, the runner-up bidder may not converge to bidding truthfully. Our first result shows that this holds for *general deterministic* truthful auctions. We also show that the ratio of the learning rates of the bidders can *qualitatively* affect the convergence of the bidders. Next, we consider the problem of revenue maximization in this environment. In the setting with fully rational bidders, Myerson (1981) showed that revenue can be maximized by using a second-price auction with reserves. We show that, in stark contrast, in our setting with learning bidders, *randomized* auctions can have strictly better revenue guarantees than second-price auctions with reserves, when $T$ is large enough. Finally, we study revenue maximization in the non-asymptotic regime. We define a notion of *auctioneer regret* comparing the revenue generated to the revenue of a second price auction with truthful bids. When the auctioneer has to use the same auction throughout the interaction, we show an (almost) tight regret bound of $\widetilde{\Theta}(T^{3/4})$. If the auctioneer can change auctions during the interaction, but in a way that is oblivious to the bids, we show an (almost) tight bound of $\widetilde{\Theta}(\sqrt{T})$.

## 1 Introduction

In auction design, truthfulness is a highly sought-after property. It allows bidders to simply reveal their true valuations, simplifying the bidding process. In the standard single item setting with fully rational profit-maximizing bidders, Myerson's seminal paper Myerson (1981) shows that an auctioneer can achieve optimal revenue by using a *truthful and deterministic* auction mechanism – a Second Price Auction (SPA) with a reserve price.

In many applications nowadays, buyers no longer bid directly in the auction but, instead, use learning algorithms to bid on their behalf. For example, in online advertising, platforms offer automated bidding tools that manage ad campaigns on behalf of advertisers. Such bidders learn to bid over many rounds and are not fully rational. In a surprising result, Kolumbus and Nisan (2022a) show that some appealing properties of second-price auctions break down in the presence of such learning bidders. In particular, when (profit-maximizing) bidders use no-regret learning algorithms, the second-price auction does not achieve as much revenue as with fully rational bidders. Indeed, bidders do not learn to bid their value, and consequently, the runner-up bidder's bid is less than their value with positive probability, which diminishes the second price auction's revenue. Moreover, Kolumbus and Nisan

---

[*]Part of the work was done while the author was a research intern at Google Research in Mountain View.

38th Conference on Neural Information Processing Systems (NeurIPS 2024).

(2022b) show that for a setting where rational agents are using learning algorithms to bid, then it is no longer optimal to truthfully submit their value as the input to the learning algorithm. This raises a crucial question: are there truthful auctions that promote convergence to the true valuations within a learning environment, and can they also guarantee strong revenue performance?

In this paper we provide an affirmative answer to this question. In doing so, we also showcase the value of *randomized* mechanisms — often overlooked in settings with profit-maximizing bidders — for environments where bidders are learning agents. While randomization introduces inherent inefficiencies due to allocations to low-valuation bidders, this very behavior facilitates learning among low-valuation bidders. A revenue-maximizing auctioneer must now carefully balance the randomization within a truthful mechanism to incentivize learning without incurring excessive revenue loss due to mis-allocation.

We build our theory based on the model presented by Kolumbus and Nisan (2022a). We consider single-item repeated interactions over $T$ periods. There are two profit-maximizing bidders participating in the auctions, with valuations that are drawn independently from the same distribution, and fully persistent over time. This assumption is motivated by online ad auctions, where multiple auctions are taking place every second, and the valuations of the advertisers remain stable for certain time scales, e.g., a day or a week. Thus, there is typically a very large sequence of auctions where the valuations of the participating agents are persistent. Bidders use mean-based no-regret learning algorithms (Braverman et al., 2018) and receive full feedback on which they base their updates. (Many of our results extend immediately to multiple bidders. We discuss other extensions, such as the partial feedback settings, in Appendix G.) The auctioneer focuses on truthful auctions, and their objective is to maximize the total revenue they achieve over the $T$ rounds of interaction. Our results are the following:

## 1.1 Our Results and Techniques

**Limitations of Deterministic Auctions.** Our first set of results (in Section 3) characterize the convergence of learners who are using Multiplicative Weights Update (MWU) in repeated *deterministic auctions*. In particular, we show the following sharp phase transition:

- If the *learning rate* of the winning type is at least as fast as the learning rate of the runner-up type, then the runner-up type will not converge to bidding truthfully, even as $T \to \infty$; in fact, it will be bidding strictly below its true value, in expectation.
- On the other hand, we show that in many auctions, such as SPA, if the learning rate of the runner-up type is strictly faster than that of the winning type, then the runner-up type will indeed converge to truthful bidding.

These generalize the results of Kolumbus and Nisan (2022a) who showed that in SPA, when bidders are using MWU with the same learning rate, then the low type will not converge to bidding truthfully. The main challenges to proving this set of results arise from our study of general deterministic auctions, which have less structure than second-price auctions. Indeed, small differences in the learning rates can affect the landscape qualitatively, as is manifested from our results. Moreover, while the auctions are deterministic, the learning algorithms are randomized and highly correlated. Hence our approach is to break down the interaction into several epochs and establish some qualitative properties which hold, with high probability, at the end of each epoch. This requires a careful accounting of the cumulative utility of each bid of both bidders within every epoch; in particular, if our estimation is off by even some $\omega(1)$ term, then it will not be sufficient to establish our result.

**Strictly-IC Auctions and the Power of Randomized Mechanisms.** The results in Section 3 show that since the low valuation bidder tends to underbid, an auctioneer using SPA with reserve makes strictly less revenue than that predicted by the model with rational agents. Motivated by this, we consider a special class of randomized auctions called *strictly-IC auctions*. These are randomized truthful auctions where for each bidder, it is strictly better to bid their true valuation compared to any other bid. We show that any strictly-IC auction is asymptotically truthful: that is, the limit point of the bidder's bid converges to their true value. Furthermore, we provide a black-box transformation from any truthful auction $A$ (deterministic or not) to a randomized auction $A'$ that has the following two properties: (i) the bidders converge towards truthful bidding, and (ii) the difference between the allocation and payment rules of the original auction $A$ and its strictly-IC counterpart $A'$ are negligible for any bid profile. Hence, such an auction $A'$ behaves similarly to $A$, but, crucially, it conveys

information to the low bidder to help it converge to truthful bidding. As a corollary of this result, we get that SPA with reserve is not revenue-maximizing in this setting, and that randomization can get strictly more revenue than SPA with reserve. This is in stark contrast with the seminal result of Myerson (1981) which shows that SPA with reserve is optimal for rational bidders.

At a more conceptual level, our results for randomized mechanisms can be viewed as showing that having enough randomness is key to the low bidder converging to truthful bidding: this randomness can come from the process itself, e.g., if bidder values are independently drawn in each round, as in Feng et al. (2021). But if not, and if the ranking of the bidders does not change much due to the lack of inherent randomness, our results show that injecting external randomness into the auction induces the desired learning behavior and hence improves the revenue. Having persistent valuations is just one case of the ranking of the bidders remaining stable over time: studying this case allows us to showcase our main ideas, but a central message of our work is that the presence or absence of stability in the rankings of the bidders is the main factor that dictates convergence to truthful bidding.

**A Non-Asymptotic Analysis.** Our next set of results in Section 5 address the non-asymptotic regime. Here we consider the *prior-free* setting, meaning that the valuations of the bidders could be drawn from potentially different distributions that are unknown to the auctioneer. In order to evaluate its revenue performance when bidders are learning agents, we introduce the notion of *auctioneer regret* for an auction, which measures the difference between the revenue achieved over $T$ rounds of implementing a given auction with learning bidders and the revenue achieved by implementing the optimal auction with rational bidders (i.e., SPA with a reserve price). Proposition 5.2 shows that if the auctioneer is constrained to use the same auction rule for all $T$ rounds, then no truthful auction — deterministic or randomized — can achieve an auctioneer-regret better than $\widetilde{O}(T^{3/4})$ in the setting of adversarial valuations. However, if the auctioneer can change the auction rule just once within the $T$ rounds, with the change happening at a time independent of the bid history, then the auctioneer's regret drops to $\widetilde{O}(\sqrt{T})$, as we show in Section 5 Moreover, we show in Proposition 5.4 that this bound of $\widetilde{O}(\sqrt{T})$ is optimal even if the auctioneer can design the auction schedule. As a byproduct of our result, we show that the first-stage randomized auction used by the mechanism leads to the fastest convergence to truthful bidding from no-regret learning agents.

To show that an auctioneer facing learning bidders using MWU must suffer an $\Omega(T^{3/4})$ revenue loss compared to the setting when it is facing rational agents, we break down the revenue loss into two non-overlapping epochs: one where the learning bidders have not converged to truthful bidding, and the other where the bidders are truthful. Now an auctioneer using the same auction throughout the interaction faces a trade-off: they can speed up the learning process to reduce the revenue loss from the first epoch, but this loses revenue in the second epoch due to the fact that the auction now differs significantly from SPA. Our result optimizes this trade-off to show that an $\Omega(T^{3/4})$ revenue loss is unavoidable. This naturally suggests decomposing the interaction into two epochs: in the first one, the auctioneer uses a truthful auction to facilitate the convergence to truthful bidding, and in the second one it uses SPA. We then design an auction that guarantees the fastest convergence to truthful bidding for mean-based learners in the prior-free setting, and we show that an improved revenue loss of at most $\widetilde{O}(\sqrt{T})$ can be achieved with this approach. (Importantly, to maintain truthfulness, the decisions of the auctioneers are fixed before the beginning of the interaction and are not affected by the bids.) This regret of $\widetilde{O}(\sqrt{T})$ seems surprising, because in traditional no-regret learning settings the optimal regret is achieved when the *exploration* and *exploitation* phase are intermixed.

## 1.2 Related Work

The most closely related works to our setting are Feng et al. (2021); Deng et al. (2022); Kolumbus and Nisan (2022a); Banchio and Skrzypacz (2022); Rawat (2023). All these works study the long-term behavior of bidding algorithms that participate in repeated auctions, focusing on first-price and second-price auctions, but they give qualitatively different results. This is because they make different assumptions across two important axes: the type of learning algorithms that the bidders use and whether their valuation is *persistent* across the interaction or it is freshly drawn in each round. Feng et al. (2021) studied the convergence of no-regret learning algorithms that bid repeatedly in second-price and first-price auctions, where all agents have i.i.d. valuation that are redrawn in every round from a discrete distribution that has non-negligible mass on each point. They show that in this setting the bidders exhibit the same-long term behavior in both second-price and first-price

auctions that classical theory predicts, i.e., the bids in second-price auctions are truthful and the bids in first-price auctions form Bayes-Nash equilibria. Kolumbus and Nisan (2022a) studied the same setting with the crucial difference that agents' valuations are persistent across the execution and they are not resampled from some distribution at every iteration. Interestingly, they showed that in the case of two bidders with in second-price auctions, the agent that has the highest valuation will end up bidding between the low valuation and its valuation, whereas the agent with the low type will end up bidding strictly below its valuation. Intuitively, in their setting the high type bidder quickly learns to bid above the valuation of the low type bidder and always win the auction, and thus the low type does not get enough signal to push its bid distribution up to its valuation. On the other hand, when the valuations are redrawn as in Feng et al. (2021), the competition that the agents face varies. In the long run, this gives enough information to the algorithms to realize that bidding truthfully is the optimal strategy. In the case of first-price auctions where the agents have persistent valuations, both Kolumbus and Nisan (2022a); Deng et al. (2022) provide convergence guarantees of no-regret learning algorithms. The type of "meta-games" we touch upon in our work, where we want to understand the incentives of the agents who are submitting their valuations to bidding algorithms that participate in the auctions on the behalf of these agents, were originally studied by Kolumbus and Nisan (2022a) and, subsequently, for more general classes of games by Kolumbus and Nisan (2022b).

The pioneering work of Hart and Mas-Colell (2000) showed that when players deploy no-regret algorithms to participate in games they converge to *coarse-correlated equilibria*. Recently, there has been a growing interest in the study of no-regret learning in repeated auctions. The empirical study of Nekipelov et al. (2015) showed that the bidding behavior of advertisers on Bing is consistent with the use of no-regret learning algorithms that bid on their behalf. Subsequently, Braverman et al. (2018) showed, among other things, that when a seller faces a no-regret buyer in repeated auctions and can use non-truthful, it can extract the whole welfare as its revenue. A very recent work (Cai et al., 2023) extended some of the previous results to the setting with multiple agents. For a detailed comparison between our work and Cai et al. (2023), we refer to Appendix B.

Banchio and Skrzypacz (2022); Rawat (2023) diverge from the previous works and consider agents that use $Q$-learning algorithms instead of no-regret learning algorithms. Their experimental findings show that in first-price auctions, such algorithmic bidders exhibit collusive phenomena, whereas they converge to truthful bidding in second-price auctions. One of the main reasons for these phenomena is the *asynchronous* update used by the $Q$-learning algorithm. The collusive behavior of such algorithms has also been exhibited in other settings (Calvano et al., 2020; Asker et al., 2021, 2022b; den Boer et al., 2022; Epivent and Lambin, 2022; Asker et al., 2022a). Notably, Bertrand et al. (2023) formally proved that $Q$-learners do collude when deployed in repeated prisoner's dilemma games.

In a related line of work, Zhang et al. (2023) study the problem of steering no-regret learning agents to a particular equilibrium. They show that the auctioneer can use *payments* to incentivize the algorithms to converge to a particular equilibrium that the designer wants them to. An interpretation of our results is that *randomization* is a way to achieve some kind of equilibrium steering in repeated auctions.

Diverging slightly from the setting we consider, some recent papers have illustrated different advantages of using randomized auctions over deterministic ones. Mehta (2022); Liaw et al. (2023) showed that there are randomized auctions which induce equilibria with better welfare guarantees for value-maximizing autobidding agents compared to deterministic ones. In the setting of revenue maximization in the presence of heterogeneous rational buyers, Guruganesh et al. (2022) showed that randomization helps when designing prior-free auctions with strong revenue guarantees, when the valuations of the buyers are drawn independently from, potentially, non-identical distributions.

## 2 Model

Our model follows the setup used in Kolumbus and Nisan (2022a). There are $T$ rounds, and the auctioneer sells a single item in each round $t = 1, \ldots, T$. There are two bidders, with bidder $i \in \{1, 2\}$ having a persistent private valuation $v_i$ drawn i.i.d. over the discrete set $B_\Delta := \{0, 1/\Delta, 2/\Delta, \ldots, 1\}$ from a regular distribution $F$. (A discrete distribution is *regular* if the discrete virtual valuation function $\phi(v) := v - \frac{1}{\Delta} \frac{\sum_{v' \geq v} \mathbf{Pr}[v']}{\mathbf{Pr}[v]}$ is non-decreasing.) Given an allocation probability $x$ and price $p$, the bidder with valuation $v$ receives a payoff of $v \cdot x - p$. In what follows, we refer to the bidder with valuation $v_L = \min\{v_1, v_2\}$ (resp. $v_H = \max\{v_1, v_2\}$) as the *low type* (resp. *high type*).

We are interested in truthful auctions, (also called strategy-proof auctions, or dominant-strategy incentive-compatible mechanisms) that are individually rational, so that at every round $t$ the auctioneer uses a mechanism $((x_1^t, x_2^t), (p_1^t, p_2^t))$ satisfying

$$v_i \cdot x_i^t(v_i, b') - p_i^t(v_i, b') \geq v_i \cdot x_i^t(b, b') - p_i^t(b, b'), \qquad \forall v_i, b, b' \in B_\Delta, \, i = 1, 2,$$
$$v_i \cdot x_i^t(v_i, b') - p_i^t(v_i, b') \geq 0, \qquad \forall v_i, b' \in B_\Delta, \, i = 1, 2.$$

In this work, we study various properties of *randomized* truthful auctions.

**Definition 2.1** (Randomized Truthful Auction). *A truthful auction $((x_1, x_2), (p_1, p_2))$ is randomized if there is some bid profile $(b_1, b_2) \in B_\Delta$ such that either $x_1(b_1, b_2) \in (0, 1)$ or $x_2(b_1, b_2) \in (0, 1)$.*

Bidders employ *learning algorithms* that bid over the $T$ rounds. We assume that the learning algorithms are *mean-based no-regret* learning algorithms (Braverman et al., 2018). For the following discussion, define $U_i^t(b \mid \mathbf{b}_{-i}^t) := \sum_{\tau=1}^t v_i \cdot x_i^\tau(b, b_{-i}^\tau) - p_i^\tau(b, b_{-i}^\tau)$ to be the cumulative reward of agent $i$ when they bid $b$ over the $t$ rounds, whereas the other agent's bids are $\mathbf{b}_{-i}^t = \{b_{-i}^\tau\}_{\tau \in [t]}$. The mean-based property states that if a bid $b \in B_\Delta$ has performed significantly better than bid $b' \in B_\Delta$, then the probability of bidding $b'$ in the next round is negligible. This is formalized below.

**Definition 2.2** (Mean-Based Property (Braverman et al., 2018)). *An algorithm for agent $i$ is $\delta$-mean-based if for any bid sequence $\mathbf{b}_{-i}^t$ such that $U_i^{t-1}(b \mid \mathbf{b}_{-i}^t) - U_i^{t-1}(b' \mid \mathbf{b}_{-i}^t) > \delta \cdot T$, for some $b, b' \in B_\Delta$, the probability of playing bid $b'$ in the next round is at most $\delta$. We say that an algorithm is mean-based if it is $\delta$-mean-based for some $\delta = o(1)$.*

The no-regret learning property states that the cumulative utility that the bidding algorithm generates is close to the cumulative utility that the optimal fixed bid would have generated, regardless of the history of bids the other bidders played. This is formalized in Definition C.1. Mean-based no-regret learning algorithms are becoming a standard class of learning algorithms to use in auction environments (see, e.g., Braverman et al. (2018); Feng et al. (2021); Deng et al. (2022); Kolumbus and Nisan (2022a), and references therein) and include many known no-regret learning algorithms, including the multiplicative-weights update algorithm (MWU). For completeness, we present the version of MWU that we use in our work in Algorithm 1. The above definitions consider a fixed value of $T$. Thus, given a sequence of such values $T$ and the limiting behavior as $T \to \infty$, we say that a family of algorithms, parameterized by the time horizon $T$, satisfies the mean-based definition if there exists $\{\delta_T\}_{T \in \mathbb{N}}$ such that $\delta_T \to_{T \to \infty} 0$, and each algorithm in this family is $\delta_T$-mean-based. We define the no-regret property of such a family of algorithms in a similar way. In general, the asymptotic behavior of the algorithms we study in this work is with respect to $T$ and the big $O$ notation suppresses quantities that do not depend on $T$.

For the sake of exposition, we focus on the *full feedback* setting: after every round $t \in [T]$, the algorithm learns for each bid $b \in B_\Delta$ the (expected) utility it would have generated had it played bid $b$. In Appendix G, we discuss potential extensions.

Throughout this paper we make a natural assumption on the algorithms which restrict bidders to never bid over their value. Specifically, for any round $t$, and any history of bids before period $t$, agent $i$ bids $b_i > v_i$ with zero probability. Without this assumption, Braverman et al. (2018); Cai et al. (2023) show that the auctioneer can extract the entire welfare in the setting where the valuations of the agents are drawn i.i.d. in each round. We focus on the *last-iterate* convergence of the distribution of the bids of the algorithms as $T \to \infty$. This is a desirable property of algorithms in multi-agent games, and recent work has focused on establishing it for learning algorithms (Cai et al., 2022b,a; Cai and Zheng, 2022). This is formalized in Definition C.2.

Due to space limitations, all the proofs of our results can be found in the appendix.

## 3 Deterministic Truthful Auctions

In this section we study the effect of the learning rate on the convergence of no-regret learning algorithms in non-degenerate *deterministic* truthful auctions. Informally, the non-degeneracy requirement states that **i)** the winning bidder $W$ under truthful bidding gets strictly positive utility, **ii)** there is some sufficiently small bid of the winning bidder such that the runner-up bidder $R$ wins the item by bidding $v_R$ but does not win by bidding $v_R - 1/\Delta$. The formal definition is given in Definition D.1.

We focus our attention to bidders that use MWU to participate in the auctions and we study the bidding distribution they converge to as a function of the ratio of the learning rate of their algorithms. Throughout this section we refer to the bidder who wins the auction under truthful bidding as the winning bidder and to the bidder that loses the auction under truthtelling as the runner-up bidder. Our main result in this section shows the following behavior in non-degenerate deterministic truthful auctions:

- The winning bidder converges to bidding between its minimum winning bid and its true value, no matter what the choice of the learning rates of the algorithms are.

- If the learning rate of the runner-up bidder is strictly faster than the learning rate of the winning bidder, then the runner-up bidder converges to bidding truthfully.

- If the learning rate of the runner-up bidder is not strictly faster than that of the winning bidder, then the runner-up bidder converges to a bidding distribution whose mean is strictly smaller than its true value. This result holds under an even milder requirement than non-degeneracy. Namely, as long as the utility of the winning bidder under truthful bidding is strictly positive.

We remark that, when the learning rates of the algorithms are instantiated before the random draw of the two valuations of the agents that are i.i.d. from some distribution $F$, then with probability at least $1/2$ the runner-up bidder will not converge to bidding truthfully, if the underlying auction is deterministic. As we will show later, this behavior worsens the revenue guarantees of the auction.

Let us first set up some notation to facilitate our discussion. We denote by $v_W \in \{v_L, v_H\}$ and $\eta_T^W$ (resp., $v_R \in \{v_L, v_H\}$, and $\eta_T^R$) the value and learning rate of the winning bidder (i.e., the one who wins if both bidders bid truthfully) and the runner-up bidder, respectively. We would like to remind the readers that, typically, the learning rate $\eta_T$ of MWU is a decreasing function of $T$ and is chosen in a way to minimize the quantity $C_\Delta/\eta_T + C'_\Delta \cdot \eta_T \cdot T$, where $C_\Delta, C'_\Delta$ are discretization-dependent constants. Usually, it is instantiated with $\eta_T = 1/\sqrt{T}$. However, for the purposes of our analysis we will say that $\eta_T$ is *non-degenerate* if $\lim_{T \to \infty} \eta_T \cdot T = \infty, \lim_{T \to \infty} \eta_T \cdot \log T = 0$. The intuition is that if the learning rate is slower than $1/T$, the bidder will be adjusting its bid distribution very slowly, so it will not learn to bid correctly. On the other hand, if the rate is faster than $1/\log T$ then the bidder will be adjusting its distribution too aggressively.

Our results show that in *deterministic* auctions the convergence behavior of the bidders depends heavily on the ratio between the learning rates. In particular, for the bidder with valuation $v_W$, we show that its bids converge to a distribution supported between $\hat{p}$, the price it would pay if both bidders bid truthfully, and its value $v_W$, no matter what the choice of the learning rate of its algorithm is. On the other hand, the convergence behavior of the runner-up bidder is more nuanced: if $\eta_T^R/\eta_T^W = \omega(1)$, i.e., the runner-up bidder learns more aggressively than the winning bidder, then it converges to bidding truthfully. However, if $\eta_T^R/\eta_T^W < C_\Delta$, where $C_\Delta$ is some discretization-dependent constant, then the runner-up converges to a bidding distribution that puts positive mass on every (discretized) point between $0$ and $v_R$, and, in particular, its expected value is strictly less than $v_R$. We remark that even though our proof idea is inspired by Kolumbus and Nisan (2022a), our analysis considers all the possible learning rates that MWU could be instantiated with and requires a more technically involved argument. In particular, we notice that while the result of Kolumbus and Nisan (2022a) is, implicitly, proved for *identical* learning rates, we show that the choice of the learning rate affects the qualitative behavior of the algorithms in a crucial way.

We prove this result in two parts. We start with the case where $\eta_T^R/\eta_T^W < C_\Delta$. The idea of the proof is to split the horizon into consecutive periods of size $O(1/\eta_T^R)$, which we call *epochs*. Now following the idea of Kolumbus and Nisan (2022a), we show that within each epoch the runner-up bidder bids truthfully $\Omega(1/\eta_T^W)$ many times, so the total utility of the winning bidder for bidding between $\hat{p}$ and $v_W$ will be at least $\Omega(1/\eta_T^W)$ greater than bidding anything between $0$ and $\hat{p} - 1/\Delta$. Because its learning rate is $\eta_T^W$, this means that it will move a constant fraction of its mass from the region $\{0, 1/\Delta, \ldots, \hat{p} - 1/\Delta\}$ to the region $\{\hat{p}, \ldots, v_W\}$. Summing this geometric series, we see that the winning bidder will submit bids in the region $\{0, 1/\Delta, \ldots, \hat{p} - 1/\Delta\}$ at most $O(1/\eta_T^W)$ many times. Let us now focus on the runner-up bidder. Following the previous argument, its total utility for bidding $v_R$ will be at most $O(1/\eta_T^W)$ greater than bidding some other bid $b' \in B_\Delta$. Since $\eta_W^R/\eta_T^W < C$, this means the probability of bidding $b'$ after $T$ rounds is only smaller than the probability of bidding

$v_R$ by a discretization-dependent multiplicative constant. The formal statement of this result and its proof follow are postponed to Appendix D.

Our next result illustrates that the convergence behavior of the runner-up type exhibits a sharp phase-transition phenomenon: if $\eta_T^R$ is even slightly faster than $\eta_T^W$, i.e., $\eta_T^R/\eta_T^W = \omega(1)$, then the runner-up will learn to bid truthfully. Let us first give a high-level idea of the proof. Similarly as before, we split the horizon into intervals of size $O(1/\eta_T^W)$. We consider the first interval of this interaction. Because of the choice of the learning rate, we can show that the winning bidder will bid $v_R - 1/\Delta$ at least $\Omega(1/\eta_T^W)$ many times. Thus, this means that the total utility of bidding $v_R$ for the runner-up bidder will be at least $\Omega(1/\eta_T^W)$ greater than bidding any other bid. Since $\eta_T^R/\eta_T^W = \omega(1)$, after the first epoch the MWU algorithm will place all but a $o(1)$-fraction of the probability mass to bidding truthfully. The formal statement and its proof appear in Appendix D.

Next, we discuss the implications that our results have to the revenue guarantees of the auctioneer. In the setting with rational bidders, the seminal work of Myerson (1981) showed that using second-price auctions with an anonymous reserve price, which depends on the value distribution $F$, generates the optimal revenue for the auctioneer. Our next result shows that this is no longer true when the bidders are learning agents, even when the valuations of the agents are drawn i.i.d. from the uniform distribution on $B_\Delta$, which we denote by $U[B_\Delta]$. Intuitively, this happens because, no matter what the reserve price is, with some non-zero probability the valuations of both agents will be higher than the reserve price. Then, since the runner-up bids will be strictly lower than the true valuation, the generated revenue will be strictly lower than in the setting with rational agents, even when $T \to \infty$.

**Theorem 3.1** (SPA with Reserve Is Not Revenue Optimal). *Let two agents draw their valuations from the uniform distribution over $U[B_\Delta]$ and participate in $T$ repeated auctions using mean-based learners. Let $b_1^T, b_2^T$ be the bid distributions after $T$ rounds. Let $\mathrm{Rev}(b_1, b_2; r)$ denote the revenue of the second-price auction with reserve price $r$ when the bids are $b_1, b_2 \in B_\Delta^2$. Then, for all $r < 1 - 1/\Delta$,*

$$\mathop{\mathbf{E}}_{v_1, v_2 \sim U[B_\Delta]} \left[ \lim_{T \to \infty} \mathop{\mathbf{E}}_{b_1 \sim b_1^T, b_2 \sim b_2^T} [\mathrm{Rev}(b_1, b_2; r) \mid v_1, v_2] \right] < \mathop{\mathbf{E}}_{v_1, v_2 \sim U[B_\Delta]} [\mathrm{Rev}(v_1, v_2; r)] - c \,,$$

*where $c > 0$ is some constant that does not depend on $T$.*

# 4 The Value of Randomized Truthful Auctions: The Asymptotic Case

In this section we show that there is a class of randomized auctions such that when mean-based no-regret learners participate in them repeatedly, they converge to *truthful* bidding. This holds for any choice of the learning rates of these algorithms, which is in contrast to the results of Section 3. We start by defining a class of auctions called *strictly* IC.

**Definition 4.1** (Strictly IC Auctions). *An auction is called strictly IC if for every bidder $i \in [n]$, valuation $v_i \in B_\Delta$, and bid profile $b_{-i} \in B_\Delta^{n-1}$ it holds that $v_i \cdot x_i(v_i, b_{-i}) - p_i(v_i, b_{-i}) > v_i \cdot x_i(b, b_{-i}) - p_i(b, b_{-i}), \forall b \neq v_i$.*

The next result, which is very useful for our derivation, states that when mean-based no-regret learning algorithms bid in some strictly IC auction they converge to bidding truthfully. Recall the definition of a mean-based learner (cf. Definition 2.2) which states that if the cumulative utility of some bid $b$ up until round $t - 1$ is much smaller than the utility of some other bid $b'$, then the probability of playing $b$ in the next round $t$ is negligible. The proof is postponed to Appendix E.

**Lemma 4.2** (Convergence in Strictly IC Auctions). *Consider $n$ bidders that participate in a repeated strictly IC auction $A$ using mean-based no-regret learning algorithms. Then, as $T \to \infty$, the bidders converge to truthful bidding in a last-iterate sense.*

The next important observation is that when we are taking a non-trivial combination of an IC auction with a strictly IC auction, the resulting auction is strictly IC. The notion of mixture we consider is formalized in Definition 4.3.

**Definition 4.3** (Mixture of Auctions). *Let $A = (x(\cdot), p(\cdot))$ be an IC auction and $A' = (x'(\cdot), p'(\cdot))$ be a strictly IC auction. For some $q \in (0, 1)$ we define the q-mixture of the auctions $A, A'$ to the be auction $\widetilde{A}_q = (q \cdot x(\cdot) + (1 - q) \cdot x'(\cdot), q \cdot p(\cdot) + (1 - q) \cdot p'(\cdot))$.*

Notice that for the allocation rule $q \cdot x(\cdot) + (1-q) \cdot x'(\cdot)$ Myerson's lemma states that the corresponding payment rule that makes the auction truthful is indeed $q \cdot p(\cdot) + (1-q) \cdot p'(\cdot)$. The following claim, whose proof follows from the definition of this class of auctions, formalizes the fact that the class of strictly IC auctions is closed under mixtures with IC auctions.

**Claim 1** (Mixture of IC and Strictly IC Auction). *Let $A$, $A'$ be an IC, strictly IC auction, respectively. Then, for any $q \in (0,1)$ the auction $q \cdot A + (1-q) \cdot A'$ is strictly IC.*

We remark that we can construct strictly IC auctions using randomization; such an example is presented in Section 5. Equipped with the above results, we can show that there is a black-box transformation from any IC auction $A$ to a strictly IC auction $A'$ so that as $T \to \infty$, any mean-based learning algorithms converges to truthful bidding, and the auction $A'$ is close to the auction $A$ in the sense that $|x_i(b) - x'_i(b)| = o(1), |p_i(b) - p'_i(b)| = o(1), \forall i \in [n], \forall b \in B^n_\Delta$. The formal statement of the result follows.

**Theorem 4.4.** *Let $A$ be an IC auction for $n$ agents with valuations $v_1, \ldots, v_n$. Let each agent $i \in [n]$ use a mean-based no-regret learning algorithm to bid in the auction. Then, there exists an auction $A'$ such that for each agent $i \in [n]$ we have that $\lim_{T \to \infty} b_i^T = v_i$ and $|x_i(b) - x'_i(b)| = o(1), |p_i(b) - p'_i(b)| = o(1), \forall b \in B^n_\Delta$, where $x_i(\cdot), x'_i(\cdot)$ (resp. $p_i(\cdot), p'_i(\cdot)$) is the allocation (resp. payment) rule of $A, A'$.*

**Equilibria of Meta-Game in Repeated Strictly IC Auctions** We now describe the implications that our results have for the meta-game that we alluded to in Section 1. Recall that this game is defined as follows: the agents submit their valuations to mean-based no-regret learning algorithms and then, given these fixed valuations, they bid on the behalf of the agents in a repeated truthful auction $A$. The main question we are interested in understanding is given the specification of the auctions and the valuations of the agents, what is the optimal value they should submit to the algorithms in order to maximize their utility, after a large number of steps?

Despite the fact that $A$ is IC and IR, Kolumbus and Nisan (2022a) showed that, rather surprisingly, when two agents use MWU to participate in repeated second price auctions there are instances where the agent with the low valuation has an incentive to report a higher value to its algorithm than its true one. This is because the valuation reported by one agent affects the bidding distribution that the other agent will converge to. To illustrate this point, assume that the low type reports $v'_L > v_H$ to its bidding algorithm. Then, the bidder with type $v_H$ will take the role of the low bidder in the interaction and will converge to bidding strictly below $v_H$. Now if its expected bid is also below $v_L$, this will generate strictly positive utility for its opponent. Using our previous construction from Theorem 4.4 and transforming the auction $A$ to a strictly IC auction $A'$, we can show that in the new meta-game every agent can gain at most $o(1)$ more utility in the long run by misreporting to the algorithm than reporting its true valuation. The reason why we observe a qualitatively different behavior in our construction is that every algorithm converges to bidding its reported value, no matter what the reported values of the other agents are. Due to space constraints, we refer the interested reader to Appendix E

**Revenue Maximization in the Learning Setting** In this section, we illustrate another application of Theorem 4.4 to revenue maximization in the learning setting. We are interested in auctions with strong revenue guarantees when the bids are coming from the limiting distribution of the algorithms, as $T \to \infty$. This has the additional complication that not only do agents draw their valuations from the distribution $F$, but also their bids come from the limiting distribution that the algorithms converge to, as $T \to \infty$. As we have seen already, this distribution depends on the valuation reported to the algorithm, the particular mean-based algorithm that it is using, and, potentially, the reported valuations and the algorithms of the opposing bidders.

As we explained in Section 3, second price auctions with reserves have strictly worse revenue guarantees in the setting with learning bidders compared to the setting with rational bidders. Using our transformation described in Theorem 4.4 we can restore their revenue guarantees. The following result whose formal proof is deferred to Appendix E is, essentially, a corollary of Theorem 4.4. Let us denote by $\mathrm{Rev}(A; b_1, \ldots, b_n)$ the revenue of some auction $A$ and by $\mathrm{Rev}(\mathrm{Myerson}; b_1, \ldots, b_n)$ the revenue of Myerson's optimal auction for $F$, where the bid profile is $b_1, \ldots, b_n \in B^n_\Delta$.

**Corollary 4.5.** *Consider an environment with $n$ agents that draw their values i.i.d. from some regular distribution $F$ and participate in repeated single-item auctions using mean-based no-regret learning*

*algorithms. Then, there is a randomized auction A so that*

$$\mathop{\mathbf{E}}_{v_1,\ldots,v_n \sim F^n} \left[ \lim_{T \to \infty} \mathop{\mathbf{E}}_{b_1 \sim b_1^T, \ldots, b_n \sim b_n^T} [\mathrm{Rev}(A; b_1, \ldots, b_n)] \,\middle|\, v_1, \ldots, v_n \right]$$
$$\geq \mathop{\mathbf{E}}_{v_1,\ldots,v_n \sim F^n} [\mathrm{Rev}(\mathrm{Myerson}; \mathrm{v}_1, \ldots, \mathrm{v}_n)] - o(1).$$

Given the results from Theorem 3.1 and Corollary 4.5 we would like to remark the following.

**Remark 1** (Randomized Auctions vs. SPA with Reserve). *Our results illustrate that randomized auctions have strictly better revenue guarantees compared to SPA with reserve price, when the bidders are using mean-based no-regret learning algorithms. This is a property of randomized auctions that is not witnessed in the setting where the bidders are fully rational, as proven by Myerson (1981).*

## 5   Revenue Maximization in the Finite Time Horizon Setting

So far, we have focused on the asymptotic regime and we have studied the convergence of the learning bidders under various auctions. In this section, we study the *finite-horizon* setting, where our goal is to come up with auctions that have strong *revenue* guarantees for the auctioneer. We focus on the *prior-free* setting, meaning that the auctioneer does not have any distributional knowledge about the valuation of the agents. Similarly to the rest of the paper, we assume that the two buyers are using mean-based no-regret learning algorithms to participate in single-item auctions for $T$ rounds. Since we are working on the prior-free setting, it is natural to compete with the cumulative revenue of the second-price auction. The goal of the auctioneer is to choose an auction in a way that minimizes

$$\widetilde{\mathrm{Reg}}_T(A; v_L, v_H) = \sum_{t=1}^{T} \mathrm{Rev}(v_L, v_H; \mathrm{SP}) - \mathbf{E}\left[ \sum_{t=1}^{T} \mathrm{Rev}(b_L^t, b_H^t; A) \right],$$

where the expectation is taken with respect to the randomness of the learning algorithms and, potentially, the auction. We will refer to this benchmark as the *auctioneer regret*. One quantity that will be useful for the derivation of our results is the following

$$\gamma_A = \min_{i \in \{1,2\}, b_i, b_{-i}, v_i \in B_\Delta^3 : b_i \neq v_i} \{ (v_i \cdot x_i(v_i, b_{-i}) - p_i(v_i, b_{-i})) - (v_i \cdot x_i(b_i, b_{-i}) - p_i(b_i, b_{-i})) \},$$

i.e., the minimum increase in the utility by bidding truthfully instead of bidding non-truthfully in $A$.

Our first goal is to understand the dependence of the auctioneer regret on the time horizon $T$. Then, we will move on to establishing bounds with respect to the number of discretized bids $\Delta$. Our first result shows that given any strictly IC auction $A$ there exists an auction $A_T$ that achieves auctioneer regret $O\left(T \cdot \sqrt{\frac{\Delta \cdot \delta_T}{\gamma_A}}\right)$. This is formalized below and the proof is postponed to Appendix F.

**Proposition 5.1.** *There exists auction $A_T$ which is a mixture of some strictly IC auction $A$ and SPA such that, for all $v_L, v_H \in [0,1]^2$ and for all $\delta_T$-mean-based learning algorithms it holds that* $\widetilde{\mathrm{Reg}}_T(A_T; v_L, v_H) = O\left(\sqrt{\frac{\Delta \cdot \delta_T}{\gamma_A} \cdot T}\right), \forall v_L, v_H \in B_\Delta^2$.

We emphasize that for common mean-based no-regret learning algorithms such as MWU it is the case that $\delta_T = \widetilde{O}\left(1/\sqrt{T}\right)$, which implies that the auctioneer regret from Proposition 5.1 grows as $\widetilde{O}\left(T^{3/4}\right)$. Our next result complements this result by showing that even if the high-valuation bidder always bids truthfully and the low-valuation bidder uses MWU with learning rate $\Theta(1/\sqrt{T})$, no auction can achieve a better auctioneer regret than $O(T^{3/4})$.

**Proposition 5.2** (Lower Bound for Constant Auction Policies). *Consider a repeated auction environment where the high-valuation bidder bids truthfully and the low-valuation bidder uses MWU with rate $\Theta(1/\sqrt{T})$. Then, every truthful auction $A_T$ has an auctioneer regret $\widetilde{\mathrm{Reg}}_T(A_T; v_L, v_H) \geq C_\Delta \cdot T^{3/4}$, where $C_\Delta > 0$ is some constant that depends on the discretization parameter.*

The proof is postponed to Appendix F. We note that choosing the learning rate of MWU to be $1/\sqrt{T}$ gives the optimal no-regret guarantees. Other choices, such as $\eta_T = \Omega(1)$, have trivial regret bounds.

Having established the previous results for repeated auctions where the auctions remain *constant* across all the iterations, it is natural to ask whether we can get improved results when the auctioneer

is allowed to change the underlying auction, but in a way that is *oblivious* to the bids that bidders have submitted so far. In other words, the auctioneer has to commit to an *auction schedule* $\{A_1, \ldots, A_T\}$ *before* the beginning of the interaction. We extend the definition of the auctioneer regret in a natural way to allow for different auctions in every timestep and we denote $\widetilde{\text{Reg}}_T(A_1, \ldots, A_T; v_L, v_H) = \sum_{t=1}^{T} \text{Rev}(v_L, v_H; \text{SP}) - \mathbf{E}[\sum_{t=1}^{T} \text{Rev}(b_L^t, b_H^t; A_t)]$. Our next result shows that there exists an auction schedule where the auctioneer changes the underlying auction only *once* throughout the interaction so that its regret is bounded by $\widetilde{O}(\delta_T \cdot T)$. For typical choices of $\eta_T$ this translates to an auctioneer regret bounded by $\widetilde{O}(\sqrt{T})$. The main insight is that the auctioneer can split the interaction into two intervals: the first interval has size $T_0$, for some appropriately chosen $T_0 \in [T]$, where the auctioneer uses some strictly IC auction $A$ that encourages the learners to converge to bidding truthfully. Then, assuming that $T_0$ is large enough to guarantee this convergence, the auctioneer switches to using second-price auction. This is perhaps counterintuitive because in other no-regret learning settings, such as multi-armed bandits, the optimal regret bound is achieved when *exploration* and *exploitation* are happening simultaneously, whereas in our setting these two phases are separated.

**Theorem 5.3.** *There exists an auction schedule* $(A_1, \ldots, A_T)$ *in which* $A_1 = A_2 = \ldots = A_{T_0} = A$, *where* $A$ *is any strictly IC auction, and* $A_{T_0+1} = A_{T_0+2} = \ldots = A_T = \text{SP}$, *that achieves* $\widetilde{\text{Reg}}(A_1, \ldots, A_T; v_L, v_H) \leq O\left(\delta_T \cdot T \cdot \left(\frac{1}{\gamma_A} + \Delta\right)\right), \forall v_L, v_H \in B_\Delta^2$.

The formal proof of this result is postponed to Appendix F. The previous result shows that for $\eta_T = \widetilde{O}(1/\sqrt{T})$ the auctioneer regret of the auction schedule we designed is $\widetilde{O}(\sqrt{T})$. Thus, we see an $\widetilde{O}(T^{1/4})$ improvement compared to the previous setting where the auctioneer was restricted to be using the same auction across all iterations.

Next, we prove that even if the auctioneer uses a different auction in every step, our bound from Theorem 5.3 is (almost) optimal with respect to the time horizon $T$. The proof idea is that when the agents are using MWU with learning rate $\eta_T$, the signals in the first $O(1/\eta_T)$ steps are insufficient for them to move their bidding distribution to truthful bids. I.e., with at least some constant probability in every round within the first $O(1/\eta_T)$ rounds, they will not be bidding their true valuation. Importantly, our lower bound holds even in the (unrealistic) setting where the auctioneer can choose $A_1, \ldots, A_T$, conditioned on $v_L, v_H$. This is formalized below; the proof is postponed to Appendix F.

**Proposition 5.4.** *When two agents are using MWU with learning rate* $1/\sqrt{T}$ *to participate in repeated single-item auctions for all the auction schedules* $(A_1, \ldots, A_T)$ *it holds that* $\widetilde{\text{Reg}}(A_1, \ldots, A_T; v_L, v_H) = \Omega(\sqrt{T})$.

Having established the optimal dependence with respect to the time horizon $T$, we now shift our attention to understanding the dependence of the auctioneer regret on the discretization parameter $\Delta$. First, we define an auction $\bar{A}$ that satisfies $\gamma_{\bar{A}} = \Theta(1/\Delta^2)$.

**Definition 5.5** (Staircase Auction)**.** *We define the allocation rule of auction* $\bar{A}$ *in the following way: with probability* $1/2$ *select a bidder* $i \in \{1, 2\}$ *independently of their bids and then allocate to* $i$ *with probability* $b_i$. *We define the payment rule in the way that makes the auction truthful.*

A simple application of Myerson's lemma shows that $\gamma_{\bar{A}} = \Theta(1/\Delta^2)$. This is because between any two consecutive bids, i.e., bids whose distance is $1/\Delta$, the increase in the allocation is $1/2\Delta$ and the function is linear. A corollary of Theorem 5.3 shows the following bound in the auctioneer regret.

**Corollary 5.6.** *Let the bidders use a mean-based learner with* $\eta_T = \widetilde{O}(\sqrt{\log \Delta/T})$ *and the auctioneer use the schedule* $(A_1, \ldots, A_T)$ *with* $A_1 = \ldots = A_{T_0} = \bar{A}, A_{T_0+1} = A_{T_0+2} = \ldots = A_T = \text{SPA}$, *for* $T_0 = \widetilde{O}(\sqrt{T}/\Delta^2)$. *Then,* $\widetilde{\text{Reg}}(A_1, \ldots, A_T; v_L, v_H) \leq \widetilde{O}\left(\Delta^2 \sqrt{T}\right), \forall v_L, v_H \in B_\Delta^2$.

# 6 Conclusion

Our work studies the behavior of learning bidders in repeated single-item auctions, with persistent valuations. We show the limitations of deterministic mechanisms, and how nuances such as learning rates can qualitatively affect participant behavior. Moreover, we show that randomized auctions can encourage faster convergence of bidders to truthful behavior. We hope our work paves the way to better understanding of learning agents' behavior in single-parameter environments, and of the power of randomization.

## Acknowledgements

Anupam Gupta is supported in part by NSF grants CCF-1955785 and CCF-2006953. Grigoris Velegkas is supported in part by the AI Institute for Learning-Enabled Optimization at Scale (TILOS).

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

## A  Multiplicative Weights Update (MWU)

In this section we describe the version of MWU we consider in this work. Similar to Braverman et al. (2018), we are using the following version of the algorithm.

---

**ALGORITHM 1:** Multiplicative Weights Update Algorithm.

---

1: Choose $\eta_T = \sqrt{\frac{\log \Delta}{T}}$. Initialize $\Delta$ weights, letting $w_i^t$ be the value of the $i$th weight at round $t$. Initially, set all $w_i^0 = 1$, let $v$ be the valuation of the agent.
2: **for** $t = 1$ to $T$ **do**
3:     Choose bid $b_i$ with probability $p_i^t = w_i^{t-1} / \sum_j w_j^{t-1}$.
4:     **for** $j = 1$ to $K$ **do**
5:        Let $u_j^t = v \cdot x^t(b_j, b') - p^t(b_j, b')$
6:        Set $w_j^t = w_j^{t-1} \cdot e^{\eta_T u_j^t}$.
7:     **end for**
8: **end for**

---

## B  Further Related Work

We view our results and the setting in which we work as orthogonal to the setting of Cai et al. (2023). Firstly, they do not restrict themselves to truthful auctions, and for their welfare extraction results, the agents are allowed to overbid. Secondly, in their setting, redrawing valuations i.i.d. in every round helps the learning process (this was also observed by Feng et al. (2021)). Intuitively, consider two agents and SPA: for every valuation of player 1, there is some positive probability that player 2's draw is below it, hence player 1 will learn that bidding truthfully is strictly better (in expectation over the other random draw), which leads to the desired bidding behavior. In such a system, randomness is already present due to the draws of the valuations, which helps the convergence to the right bidding behavior.

Our work also differs from Cai et al. (2023) in having different conceptual goals: we aim to "restore" the single-shot behavior in natural auctions, such as second-price auctions, in the presence of mean-based learning agents by making minimal modifications to the underlying auction rule. On the other hand, Cai et al. (2023) aim to exploit the mean-based learning behavior to extract more revenue, and their auctions diverge from the truthful ones we consider in our work. Thus, in our setting, it is clear that reporting the valuation truthfully to the bidding algorithm is an (almost) optimal strategy for the agents (i.e., the so-called "meta-game" considered by Kolumbus and Nisan (2022a) is truthful), whereas it is not clear to us whether reporting the valuations truthfully to the no-regret algorithms is an optimal strategy in the setting of Cai et al. (2023).

## C  Omitted Details from Section 2

Skreta (2006) shows that our discrete-type space mechanism design problem approximates the mechanism design problem with continuous type space as $\Delta \to \infty$: specifically, Proposition 1 from that paper gives the following claims.

**Claim 2.** *A mechanism is truthful if and only for every $v_{-i}$ $x_i(v_i, v_{-i})$ is non-decreasing on $v_i$ and $p_i$ satisfy that*

$$\left| p_i(v_i, v_{-i}) - \left( v_i x_i(v_i, v_{-i}) - \int_0^{v_i} x_i(z, v_{-i}) dz \right) \right| \leq O(1/\Delta).$$

**Claim 3.** *Suppose bidders are rational agents (i.e., they maximize profits). Let $OPT$ be the revenue of the revenue-maximizing mechanism (among truthful or non-truthful) that the auctioneer can implement, and $Rev(r - SPA)$ be the revenue of a Second Price Auction with reserve $r$. Then for $r = \min\{v : \phi(v) \geq 0\}$, we have that $OPT = Rev(r - SPA)$.*

**Definition C.1** (No-Regret Learning Property). *Let $\{b_i^\tau\}_{\tau \in [T]}$ be the bid sequence submitted by agent $i$'s algorithm, and $U_i^T(\mathbf{b}^T) = \sum_{\tau=1}^{T} v_i \cdot x_i^\tau(b_i^\tau, b_{-i}^\tau) - p_i^\tau(b_i^\tau, b_{-i}^\tau)$ the total reward agent $i$*

*receives. We say that this algorithm* satisfies the no-regret property *if for any sequence $\mathbf{b}_{-i}^T$ it holds that*

$$\mathbf{E}\left[\max_{b \in B_\Delta} U_i^T(b \mid \mathbf{b}_{-i}^T) - U_i^T(\mathbf{b}^T)\right] = o(T)\,,$$

*where the expectation is taken with respect to the randomness of the algorithm.*

**Definition C.2** (Last Iterate Convergence (LIC)). *Let $\tilde{b}_i^T$ the bid distribution of bidder $i$ in the last round $T$. We say that $\tilde{b}_i^T$ converges to some distribution $\tilde{q}$ over $B_\Delta$ if $\lim_{T \to \infty} d_{\mathrm{TV}}(\tilde{b}_i^T, \tilde{q}) = o(1)$, where $d_{\mathrm{TV}} := \frac{1}{2}\left(\sum_{b \in B_\Delta} |\tilde{b}_i^T(b) - \tilde{q}(b)|\right)$ is the Total-Variation (TV) distance between $\tilde{b}_i^T$ and $\tilde{q}$.*

# D   Omitted Details from Section 3

**Definition D.1** (Non-Degenerate auctions). *A single-item auction $(x, p)$ for two agents is non-degenerate with respect to the valuation profile $(v_1, v_2)$ if there are bid profiles $b_1 \leq v_1, b_2 \leq v_2$, so that*

$$v_1 \cdot x_1(v_1, b_2) - p_1(v_1, b_2) > v_1 \cdot x_1(v_1 - 1/\Delta, b_2) - p_1(v_1, b_2) \geq 0$$
$$v_2 \cdot x_2(b_1, v_2) - p_2(b_1, v_2) > v_2 \cdot x_2(b_1, v_2 - 1/\Delta) - p_2(b_1, v_2 - 1/\Delta) \geq 0\,,$$

*and*

$$\max\left\{v_1 \cdot x_1(v_1, v_2) - p_1(v_1, v_2), v_2 \cdot x_2(v_1, v_2) - p_2(v_1, v_2)\right\} > 0\,.$$

In order to show our result, we utilize a characterization (cf. Theorem D.2) regarding the structure of truthful deterministic single-item auctions that charge non-negative payments (see, e.g., Roughgarden (2010, Thm 9.36)) for $n$ bidders.

**Theorem D.2** (Characterization of Truthful Deterministic Single-Item Auctions Roughgarden (2010)). *A single-item auction is truthful, and satisfies NPT, i.e., no payment transfers from the auctioneer to the bidders, if and only if:*

- $x_i(\cdot, v_{-i})$ *is monotone for every $i \in [n], v_{-i} \in B_\Delta^{n-1}$.*

- *For all $i \in [n], v_i \in B_\Delta, v_{-i} \in B_\Delta^{n-1}$ we have that*

$$p_i(v_i, v_{-i}) = \begin{cases} 0, & \text{if } x_i(v_i, v_{-i}) = 0 \\ \min\{b \in B_\Delta : x_i(b, v_{-i}) = 1\}, & \text{if } x_i(v_i, v_{-i}) = 1 \end{cases}\,.$$

**Theorem D.3** (No Deterministic Auction Leads to Truthful Bidding). *Fix a valuation profile $(v_1, v_2)$ and a deterministic truthful auction. Suppose bidders bid using MWU and with non-degenerate learning rates. Let $W$ (respectively $R$), be the bidder $i \in \{1, 2\}$ such that $x_i(v_i, v_{-i}) = 1$ (respectively, $x_i(v_i, v_{-i}) = 0$) and let $\hat{p} = p_W(v_W, v_R)$. Assume that $\lim_{T \to \infty} \eta_T^R/\eta_T^W < \infty$ and $v_W \cdot x_W(v_W, v_R) - \hat{p} > 0$. Then, with probability at least $0.99$, the winner's bids converge to a distribution supported between $\hat{p}, v_W$ and the runner-up bidder converges to a bidding distribution satisfying $0 < \mathbf{Pr}[0] \leq \mathbf{Pr}[1/\Delta] \leq \ldots \leq \mathbf{Pr}[v_R]$.*

*Proof of Theorem D.3.* The idea of the proof is to split the horizon $T$ into continuous non-overlapping epochs of length $c/\eta_T^W$, where $c$ is some sufficiently large constant that depends on the discretization parameter $\Delta$. Notice that since $\lim_{T \to \infty} \eta_T^W \cdot T = \infty$ these epochs are well-defined, when $T$ is sufficiently large. Assume without loss of generality that the weights of all the bids that are at most $v_W$ (resp. $v_R$) for the winning bidder (resp. runner-up) are initialized to 1. (The proof holds as long as there is some constant mass on each bid at the initialization stage, albeit with different constants.) We denote the epochs by $\tau$ and the rounds of the interaction by $t$.

Let $c_W = v_W - \hat{p}$ be the utility the bidder gets when it wins the auction. By assumption, $c_W > 0$. Let $W_W$ be the set of bids between $\hat{p}$ and $v_W$, i.e., $W_W = \{\hat{p}, \hat{p} + 1/\Delta, \ldots, v_W\}$. Whenever the runner-up bids $v_R$ all the bids in $W_W$ increase their weights by a multiplicative factor of $e^{c_W \cdot \eta_T^W}$, whereas the weights of the other bids remain unchanged. Moreover, since the allocation rule is non-decreasing and the price does not depend on the bid, whenever the weight of some bid $b \in B_\Delta$ is increased, the weights of all the bids that are greater than $b$ are also increased by the same amount. Notice that, since bidding $v_R$ is a weakly-dominant strategy for the runner-up type, the mass that it

puts on $v_R$ will never decrease relatively to the mass of the rest of the bids. Thus, the probability of bidding $v_R$ for the runner-up type is at least $1/\Delta$ in every round. Hence, if we consider an interval of size $T_0 = 8\Delta^2/(\eta_T^W \cdot c_W)$ and we denote by $Z_i, i \in [T_0]$, the indicator variable of whether the runner-up bid $v_R$ in round $i \in [T_0]$ we have that for any $\alpha > 0$

$$\mathbf{Pr}\left[Z_1 + \ldots + Z_{T_0} \geq \alpha\right] \geq \mathbf{Pr}\left[\tilde{Z}_1 + \ldots + \tilde{Z}_{T_0} \geq \alpha\right],$$

where $\tilde{Z}_i, \in [T_0]$ are i.i.d. Bernoulli random variables with mean $1/\Delta$. Then, the multiplicative version of Chernoff bound on $\{\tilde{Z}_i\}_{i \in [T_0]}$ shows that, with probability at least $1 - e^{-\Delta/(\eta_T^W \cdot c_W)}$ the runner-up type will bid at least $4\Delta/(\eta_T^W \cdot c_W))$ many times $v_R$ in this window. By a union bound, we know that with probability at least $1 - (T \cdot \eta_T^W/c) \cdot e^{-\Delta/(\eta_T^W \cdot c_W)}$ this holds across all the $T \cdot \eta_T^W/c$ different epochs. We call this event $\mathcal{E}_1$ and condition on it for the rest of the proof. Our assumption that $\eta_T$ is non-degenerate shows that this probability is at least $1 - o(1)$.

Let $w_W^\tau(b)$ be the total weight that the winning type assigns to $b$ at the beginning of epoch $\tau$ and $m_W^\tau(b)$ be its probability. Notice that at $\tau = 1$ this distribution is uniform. Consider the ratio of the weights of any $b \leq \hat{p} - 1/\Delta$ and $\hat{p}$. We have that

$$\frac{w_W^{\tau+1}(b)}{w_W^{\tau+1}(\hat{p})} \leq \frac{w_W^\tau(b)}{w_W^\tau(\hat{p})} \cdot e^{-4c_W \cdot \Delta \cdot \eta_T^W/(c_W \cdot \eta_T^W)} = \frac{w_W^\tau(b)}{w_W^\tau(\hat{p})} \cdot e^{-4\Delta}, \tag{1}$$

where $w_W^\tau(b), w_W^\tau(\hat{p})$ are the weights that the winner puts on $b, \hat{p}$ at the beginning of epoch $\tau$ (similarly for the $\tau + 1$ terms). For the probability of each bid in MWU, $m_W^{\tau+1}(b) = \frac{w_W^{\tau+1}(b)}{\sum_{b' \in B_\Delta} w_W^{\tau+1}(b')}$ (and symmetrically for the other terms). Thus, by dividing the numerator and the denominator of the RHS of Equation (1) by $\sum_{b' \in B_\Delta} w_W^\tau(b')$ and the numerator and denominator of the LHS of Equation (1) by $\sum_{b' \in B_\Delta} w_W^{\tau+1}(b')$ we get:

$$\frac{m_W^{\tau+1}(b)}{m_W^{\tau+1}(\hat{p})} \leq \frac{m_W^\tau(b)}{m_W^\tau(\hat{p})} \cdot e^{-4\Delta}.$$

Multiplying by $m_W^{\tau+1}(\hat{p})$ gives us

$$m_W^{\tau+1}(b) \leq \frac{m_W^{\tau+1}(\hat{p})}{m_W^\tau(\hat{p})} \cdot m_W^\tau(b) \cdot e^{-4\Delta}.$$

Notice that $m_W^1(\hat{p}) = 1/\Delta, m_W^\tau(\hat{p})$ is non-decreasing in $\tau$ since bidding $\hat{p}$ is a weakly-dominant strategy for the winning type[2], and, by definition, $m_H^{\tau+1}(\hat{p}) \leq 1$, so $\frac{m_W^{\tau+1}(\hat{p})}{m_W^\tau(\hat{p})} \leq \Delta$. Hence,

$$m_W^{\tau+1}(b) \leq \Delta e^{-4\Delta} \cdot m_W^{\tau+1}(b) < 0.1 \cdot m_W^\tau(b), \forall b < \hat{p},$$

where the second inequality follows from $xe^{-4x} < 1, \forall x > 0$. Thus, after each epoch the probability that the winning type does not bid in $W_W$ decreases by a factor of $0.9$. Hence, we can see that after $O(\eta_T^W \cdot T)$ epochs that total mass in this region is at most $O(0.1^{\eta_T^W \cdot T - 1}) = o(1)$. This proves the claim about the distribution of the winning type.

Let $Z_i, i \in [T]$, be the random variable that indicates whether $v_W$ bid in $\{0, 1/\Delta, \ldots, \hat{p} - 1/\Delta\}$ in round $i \in [T]$. Let also $T'$ denote the total number of epochs. Let $\widehat{Z}_\tau = Z_\tau + \ldots + Z_{\tau+T_0-1}$, so that $\mathbb{E}[Z_1 + \ldots Z_T] = \sum_{\tau=1}^{T'} \mathbb{E}[\widehat{Z}_\tau]$. The preceding steps of the proof had shown that after every round, the probability that the winner bids in this region is non-increasing (since the bids in interval $I$ are weakly dominated by the bids in $\{\hat{p}, \ldots, v_W\}$), hence $\mathbb{E}[\widehat{Z}_\tau] \leq T_0 \cdot \mathbb{E}[Z_{(\tau-1)\cdot T_0+1}]$. Thus, it suffices to bound $\sum_{\tau=1}^{T'} \mathbb{E}[Z_{(\tau-1)\cdot T_0+1}]$.

By definition, $\mathbb{E}[Z_{(\tau-1)\cdot T_0+1}] = \sum_{b<\hat{p}} m_W^{(\tau-1)\cdot T_0+1}(b)$. Now, the previous step of the proof had shown that the mass of each bid in interval $I$ drops by a factor of $0.9$ between the beginning of

---

[2]This is where we are using the assumption that the runner-up type does not overbid. Otherwise, the argument can still go through with a different constant since we can show that the winning type will overbid only some $O(\eta_T^W)$ many times and we need to account for this term.

consecutive epochs, i.e., $m_W^{\tau \cdot T_0 + 1}(b) \leq 0.1 \cdot m_W^{(\tau-1) \cdot T_0 + 1}(b)$ for all $b \in \{0, 1/\Delta, \ldots \widehat{p} - 1\}$. This implies $\mathbb{E}[Z_{\tau \cdot T_0 + 1}] \leq 0.1 \cdot \mathbb{E}[Z_{(\tau-1) \cdot T_0 + 1}]$. Using $\mathbb{E}[Z_1] \leq 1$, we get $\sum_{\tau=1}^{T'} \mathbb{E}[Z_{(\tau-1) \cdot T_0 + 1}] \leq \sum_{\tau=1}^{T'} (0.1)^{\tau-1}$. Multiplying this by the value of $T_0$ gives

$$\mathbf{E}\left[Z_1 + \ldots + Z_T\right] \leq \sum_{\tau=1}^{T'} (8\Delta^2/(\eta_T^W \cdot c_W)) \cdot (0.1)^{\tau-1}$$
$$\leq \sum_{\tau=1}^{\infty} (8\Delta^2/(\eta_T^W \cdot c_W)) \cdot (0.1)^{\tau-1}$$
$$\leq 16\Delta^2/(\eta_T^W \cdot c_W).$$

Hence, using Markov's inequality we see that

$$\mathbf{Pr}\left[Z_1 + \ldots + Z_T \geq 101 \cdot \left(16\Delta^2/(\eta_T^W \cdot c_W)\right)\right] \leq \frac{\mathbf{E}\left[Z_1 + \ldots Z_T\right]}{101 \cdot \left(16\Delta^2/(\eta_T^W \cdot c_W)\right)} \leq \frac{1}{101}.$$

Let us call this event $\mathcal{E}_2$ and condition on it.

Let us now consider the bid distribution of the runner-up type after the end of the last epoch. We denote this distribution by $\widehat{m}_R(\cdot)$. Recall that whenever the winning type bids in $W_W$, the runner-up type performs no updates. Moreover, whenever it does perform an update its utility when it bids $v_R$ is at most 1 greater than bidding $b = 0$. Notice that whenever the weight of some bid $b$ is increased, the weights of all the bids greater than $b$ are also increased by the same amount, so the monotonicity of the bid distribution follows immediately. It suffices now to bound the ratio of the probability of bidding zero and the probability of bidding $v_R$ by some quantity that is independent of $T$. We have that

$$\frac{\widehat{m}_R(0)}{\widehat{m}_R(v_R)} \geq e^{-\eta_T^R 101 \cdot \left(16\Delta^2/(\eta_T^W \cdot c_W)\right)} \implies \widehat{m}_R(0) \geq \frac{e^{-\eta_T^R 101 \cdot \left(16\Delta^2/(\eta_T^W \cdot c_W)\right)}}{\Delta},$$

where the second inequality follows from the fact that the distribution is initialized to be uniform and $v_R$ is a weakly-dominant strategy across all rounds, so its probability is not decreased. Notice that

$$\lim_{T \to \infty} \eta_T^R/\eta_T^W < C,$$

for some discretization-dependent $C$, it follows that $\widehat{m}_R(0) > C'$, where $C' > 0$ is some discretization-dependent constant. Since $\mathbf{Pr}[\mathcal{E}_1] \geq 1 - o(1), \mathbf{Pr}[\mathcal{E}_2] \geq 100/101$, we have that $\mathbf{Pr}[\mathcal{E}_1 \cap \mathcal{E}_2] \geq 99/100$, when $T$ is large enough. □

**Theorem D.4** (Effect of Learning Rate on Convergence). *Fix a valuation profile $(v_1, v_2)$ and a non-degenerate deterministic truthful auction with respect to $(v_1, v_2)$. Suppose bidders bid using MWU and with non-degenerate learning rates. Let $W$ (respectively $R$), be the bidder $i \in \{1, 2\}$ such that $x_i(v_i, v_{-i}) = 1$ (respectively, $x_i(v_i, v_{-i}) = 0$). Let $\widehat{p}$ be the minimum winning bid of $W$ when $R$ bids $v_R$. Assume that $\eta_T^R/\eta_T^W = \omega(1)$. Then, with probability at least $1 - o(1)$, bidder $R$ converges to bidding $v_R$ and bidder $W$ converges to a bidding distribution supported in $\{\widehat{p}, \widehat{p} + 1/\Delta, \ldots, v_W\}$.*

*Proof of Theorem D.4.* Consider the first $T_0 = c'_\Delta/\eta_T^W$ rounds of the game, for some $c'_\Delta$ discretization-dependent constant. Assume without loss of generality that the weights of all the bids that are at most $v_W$ (resp. $v_R$) for the winning bidder (resp. runner-up) are initialized to 1. (Again, the argument works so long as all the weights are initialized with some constants.) Since the auction is non-degenerate with respect to $v_W, v_R$, there exists some bid of the winning type $b_W \leq v_W$ so that the runner-up bidder wins the auction when bidding truthfully and gets positive utility, i.e.,

$$v_R \cdot x_R(v_R, b_W) - p_R(v_R, b_W) > 0.$$

Moreover, for all bids $b_R < v_R$ it holds

$$v_R \cdot x_R(v_R, b_W) - p_R(v_R, b_W) - (v_R \cdot x_R(b_R, b_W) - p_R(b_R, b_W)) > 0.$$

Since the auction is truthful, the difference above is minimized at $b_R = v_R - 1/\Delta$. Let

$$u'_R := v_R \cdot x_R(v_R, b_W) - p_R(v_R, b_W) - (v_R \cdot x_R(v_R - 1/\Delta, b_W) - p_R(v_R - 1/\Delta, b_W)),$$

and, by definition, $u'_R > 0$. Let us consider the winning type and look at the worst-case ratio of the probability that is placed on bids $b^t_W = b_W, b^t_W = v_W$ at the end of every round $t \in \{1, \ldots, T_0\}$. We have that

$$\frac{\mathbf{Pr}[b^t_W = b_W]}{\mathbf{Pr}[b^t_W = v_W]} \geq e^{-\eta^W_T \cdot v_W \cdot t}$$
$$\geq e^{-\eta^W_T \cdot v_W \cdot T_0}$$
$$= e^{-c'_\Delta \cdot v_W} ,$$

where the first inequality follows from the fact that bidding $v_W$ always yields at most $v_W$ utility more than bidding any other bid and the second one because $t \leq T_0$. Moreover, since $\mathbf{Pr}[b^1_W = v_W] = 1/\Delta$ and the probability that is placed on $b^t_W = v_W$ is non-decreasing across the executions (since it is a weakly-dominant strategy), we have that

$$\mathbf{Pr}[b^t_W = b_W] \geq e^{-c'_\Delta \cdot v_W}/\Delta, \forall t \in \{1, \ldots, T_0\} .$$

Let $Z^{T_0}$ denote the random variable that counts the number of times the winning type bids $b_W$ within the first $T_0$ rounds. Let $\tilde{Z}_\tau, \tau \in [T_0]$ be independent Bernoulli random variables with mean $e^{-c'_\Delta \cdot v_W}/\Delta$. Notice that, $\forall \alpha > 0$, it holds that $\mathbf{Pr}[Z^{T_0} \geq \alpha] \geq \mathbf{Pr}[\sum^{T_0}_{\tau=1} \tilde{Z}_\tau \geq \alpha]$. Moreover,

$$\mathbf{E}\left[\sum^{T_0}_{\tau=1} \tilde{Z}_\tau\right] \geq T_0 \cdot e^{-c'_\Delta \cdot v_W}/\Delta = c'_\Delta/\eta^W_T \cdot e^{-c'_\Delta \cdot v_W}/\Delta .$$

To simplify the notation, let us denote $\tilde{c}_\Delta = c'_\Delta \cdot e^{-c'_\Delta \cdot v_H}/\Delta$. Thus, a multiplicative Chernoff bound shows that, with probability at least $1 - e^{-\tilde{c}_\Delta/(8\eta^W_T)} = 1 - o(1)$, we have that $Z^{T_0} \geq \tilde{c}_\Delta/(2\eta^W_T)$. Let us call this event $E$ and condition on it.

Let us now focus on the bid distribution of the runner-up bidder after the first $T_0$ rounds. Notice that whenever the winning bidder bids $b_W$ then bidding $v_R$ yields utility at least $u'_R$ greater than bidding any other bid to the runner-up type, and in the rounds where this does not happen, bidding $v_R$ is still a weakly dominant strategy so it generates as much utility as any other bid. Thus, we have that

$$\frac{\mathbf{Pr}[b^{T_0}_R = v_R - 1/\Delta]}{\mathbf{Pr}[b^{T_0}_R = v_R]} \leq e^{-u'_R \cdot \eta^R_T \cdot Z^{T_0}}$$
$$\leq e^{-\eta^R_T \cdot \tilde{c}_\Delta/(2\eta^W_T \Delta)}$$
$$= o(1)$$

Thus, since bidding $v_R$ is a weakly dominant strategy for the runner-up this ratio is non-increasing in $t$ we can immediately see that

$$\frac{\mathbf{Pr}[b^{T_0}_R = v_R - 1/\Delta]}{\mathbf{Pr}[b^{T_0}_R = v_R]} = o(1) ,$$

which gives that

$$\mathbf{Pr}[b^T_R = v_R - 1/\Delta] = o(1) .$$

The same argument can be applied to all bids in $\{0, 1/\Delta, \ldots, v_R - 1/\Delta\}$.

For the winning type, a symmetric argument shows that since after $O(\eta^W_T)$ many rounds the runner-up type bids $v_R$ with high probability, all the bids in the region $\{\hat{v}_W, \ldots, v_W\}$ will yield utility that is larger than bidding $v_R - 1/\Delta$ by at least $1/\Delta$ (again with high probability), so after another $\omega(\eta^W_T)$ rounds its mass will be concentrated on bidding in this region. $\square$

*Proof of Theorem 3.1.* Let $\mathcal{E} = \{r < v_1\} \cap \{r < v_2\} \cap \{v_1 \neq v_2\}$. We can decompose $\mathbf{E}_{v_1,v_2 \sim U[B_\Delta]}[\mathrm{Rev}(v_1, v_2; r)]$ as:

$$\mathop{\mathbf{E}}_{v_1,v_2 \sim U[B_\Delta]}[\mathrm{Rev}(v_1, v_2; r)] = \mathop{\mathbf{E}}_{v_1,v_2 \sim U[B_\Delta]}[\mathrm{Rev}(v_1, v_2; r)|\mathcal{E}] \cdot \mathop{\mathbf{Pr}}_{v_1,v_2 \sim U[B_\Delta]}[\mathcal{E}]$$
$$+ \mathop{\mathbf{E}}_{v_1,v_2 \sim U[B_\Delta]}[\mathrm{Rev}(v_1, v_2; r)|\mathcal{E}'] \cdot \mathop{\mathbf{Pr}}_{v_1,v_2 \sim U[B_\Delta]}[\mathcal{E}'] .$$

Notice that under $\mathcal{E}'$, the revenue of the auction in the learning setting satisfies

$$\underset{v_1,v_2\sim U[B_\Delta]}{\mathbf{E}}\left[\lim_{T\to\infty}\underset{b_1\sim b_1^T,b_2\sim b_2^T}{\mathbf{E}}[\mathrm{Rev}(b_1,b_2;r)\mid v_1,v_2]\,\middle|\,\mathcal{E}'\right]\le\underset{v_1,v_2\sim U[B_\Delta]}{\mathbf{E}}[\mathrm{Rev}(v_1,v_2;r)|\,\mathcal{E}'].$$

This is because both bidders will be bidding at most their valuation, so the revenue of the auction cannot increase. Let us now focus on the first term. Under the event $\mathcal{E}$, the revenue of the auction under rational agents is $\min\{v_1,v_2\}>r$. However, in the learning setting, the runner-up bidder will be bidding strictly below their valuation in expectation, by Theorem D.3. Hence, we have that

$$\underset{v_1,v_2\sim U[B_\Delta]}{\mathbf{E}}\left[\lim_{T\to\infty}\underset{b_1\sim b_1^T,b_2\sim b_2^T}{\mathbf{E}}[\mathrm{Rev}(b_1,b_2;r)\mid v_1,v_2]\,\middle|\,\mathcal{E}\right]<\underset{v_1,v_2\sim U[B_\Delta]}{\mathbf{E}}\left[\min\{v_1,v_2\}\,\middle|\,\mathcal{E}\right]-c'$$

$$=\underset{v_1,v_2\sim U[B_\Delta]}{\mathbf{E}}[\mathrm{Rev}(v_1,v_2;r)|\,\mathcal{E}]-c'.$$

Since $\mathbf{Pr}[\mathcal{E}]>0$, the result follows by combining the two inequalities. $\qquad\square$

# E   Omitted Details from Section 4

*Proof of Lemma 4.2.* Let

$$\gamma_A=\min_{i\in[n],v\in B_\Delta,b_{-i}\in B_\Delta^{n-1},b\in B_\Delta:b\neq v}\{u_i(v,b_{-i})-u_i(b,b_{-i})\},$$

i.e., the minimum improvement in the utility that is guaranteed to every player when they switch to bidding truthfully from any non-truthful bid, no matter what their valuation and the bids of the opponents are. Notice that for any fixed auction $A$ this quantity does not depend on $T$. Moreover, since $A$ is a strictly IC auction we have that $\gamma_A>0$. Consider any round $t\in[T]$ of the interaction. For any player $i\in[n]$, we have that

$$u^t(v_i,b_{-i}^t)-u^t(b',b_{-i}^t)\ge\gamma_A,\forall b'\neq v_i,$$

no matter what the bids $b_{-i}^t$ are. Let $\delta_1,\dots,\delta_n$ be the mean-based parameters of the algorithms that the agents are using. Moreover, let $T_0=\max_{i\in[n]}\delta_i\cdot T/\gamma_A$. Notice that since $\delta_i=o(1),\forall i\in[n]$, by picking $T$ sufficiently large we have that $T_0<T$. We immediately get that, for every player $i\in[n]$

$$\sum_{t=1}^{T_0}\left(u^t(v_i,b_{-i}^t)-u^t(b',b_{-i}^t)\right)\ge\gamma_A\cdot T_0\ge\delta_i\cdot T,\forall b'\neq v_i,$$

no matter what the bid profile $b_{-i}^t$ of the other bidders in every round is. Thus, for every bidder $i\in[n]$, by taking a union bound over all bids $b\neq v_i$, we see that in round $T_0+1$ the probability of not bidding truthfully is at most $\Delta\cdot\delta_i=o(1)$. Hence, we have shown the result. $\qquad\square$

*Proof of Theorem 4.4.* Let $\delta_1,\dots,\delta_n$ be the mean-based parameters of the algorithms that the agents are using. Recall that these parameters do depend on $T$. Assume without loss of generality that $\delta_1$ is the slowest one, i.e., $\lim_{T\to\infty}\delta_i/\delta_1\le C,\forall i\in[n]$, where $C$ is some discretization-dependent constant. Let $\widetilde{A}$ be a strictly IC auction and define

$$\gamma_{\widetilde{A}}=\min_{i\in[n],v\in B_\Delta,b_{-i}\in B_\Delta^{n-1},b\in B_\Delta:b\neq v}\{\widetilde{u}_i(v,b_{-i})-\widetilde{u}_i(b,b_{-i})\}.$$

Similarly as in the previous proof, notice that $\gamma_{\widetilde{A}}$ does not depend on $T$. Consider the $q_T$-mixture of the auctions $A,\widetilde{A}$ and let us denote this auction by $A'$. Let $x,\widetilde{x},x'$ be the allocation rules of $A,\widetilde{A},A'$, respectively, and let us define the payment rules in a symmetric way. Notice that since $x'(\cdot)=q_T\widetilde{x}(\cdot)+(1-q_T)x(\cdot),p'(\cdot)=q_T\widetilde{p}(\cdot)+(1-q_T)p(\cdot)$, it follows immediately that

$$\gamma_{A'}\ge q_T\cdot\gamma_{\widetilde{A}}.$$

Moreover, notice that

$$|x'(\cdot)-x(\cdot)|\le q_T\cdot|\widetilde{x}(\cdot)-x(\cdot)|\le q_T$$
$$|p'(\cdot)-p(\cdot)|\le q_T\cdot|\widetilde{p}(\cdot)-p(\cdot)|\le q_T.$$

Let us focus on agent 1 since it is the one that has the slowest convergence. After $T_0$ rounds of the game we have that

$$\sum_{t=1}^{T_0} \left( u^t(v_1, b_{-1}^t) - u^t(b', b_{-1}^t) \right) \geq \gamma_{A'} \cdot T_0 \geq q_T \cdot \gamma_{\widetilde{A}} \cdot T_0, \forall b' \neq v_1 \,,$$

no matter what the bid profile of the rest of the bidders in every round is. Thus, in order for the mean-based guarantee of the algorithm of the first bidder to give us the desired convergence we see that we need $T_0 \geq {\delta_1 \cdot T}/{q_T \cdot \gamma_{\widetilde{A}}}$. Since $T_0 \leq T$, this places a constraint on the choice of $q_T$, namely that $q_T \geq {\delta_1}/{\gamma_{\widetilde{A}}}$. Thus, since this is the only constraint that we have on the choice of $q_T$ we see that choosing $q_T = {2\delta_1}/{\gamma_{\widetilde{A}}} = o(1)$ suffices to get the result. $\qquad\square$

*Proof of Corollary 4.5.* Let $A'$ be the output of Theorem 4.4 when the input auction is Myerson's revenue-optimal auction for $F$. For any fixed valuation profile $v \in B_\Delta^n$, for sufficiently large $T$, each bidder $i \in [n]$ will be bidding $v_i$ except with probability $o(1)$. Moreover, the payments in these two auctions differ by some $o(1)$. Thus,

$$\mathop{\mathbf{E}}_{b_1 \sim b_1^T, \ldots, b_n \sim b_n^T} \left[ \lim_{T \to \infty} \text{Rev}(A; b_1, \ldots, b_n) \right] \geq \text{Rev}(\text{Myerson}; v_1, \ldots, v_n) - o(1) \,.$$

The result follows by taking the expectation over the random draw of $v_1, \ldots, v_n$. $\qquad\square$

We present the formal result about the equilibria of the meta-game below.

**Corollary E.1** (Equilibria of Meta-Game). *Let $A$ be an IC, IR auction. Let $T$ be the number of interactions. Assume that $n$ agents use mean-based no-regret learning algorithms to bid in these repeated auctions. Then, there is an auction $A'$ such that*

- *$|x_i(b) - x_i'(b)| = o(1), |p_i(b) - p_i'(b)| = o(1), \forall i \in [n], \forall b \in B_\Delta^n$.*

- *In the meta-game that is induced by $A'$ every agent can gain at most $o(1)$ utility by misreporting its value to the bidding algorithm.*

*Proof of Corollary E.1.* Let $v_1, \ldots, v_n$ be the values of the agents and let $\hat{v}_1, \ldots, \hat{v}_n$ be the reports to the bidding algorithms. Let $A'$ be auction obtained by feeding the auction $A$ into the transformation described in Theorem 4.4. The guarantees of this result show that

- $|x_i(b) - x_i'(b)| = o(1), |p_i(b) - p_i'(b)| = o(1), \forall i \in [n] \forall b \in B_\Delta$,

- $\mathbf{Pr}[b_i^T \neq \hat{v}_i] = o(1), \forall i \in [n]$,

where $b_i^T$ is the bid of the $i$-th agent in round $T$. Thus, with high probability after a large enough number of rounds, for every agent $i \in [n]$ the algorithm is bidding the reported value $\hat{v}_i$ no matter what the other reports $\hat{v}_{-i}$ are. Since the auction $A'$ is truthful, the utility of each agent is maximized when $b_i^T = v_i$. Hence, the optimal strategy, up to $o(1)$, is to report $v_i = \hat{v}_i, \forall i \in [n]$. To be more formal, the expected utility of the $i-$th agent in round $T$ is

$$\mathbf{E}\left[ u_i'(b_i^T, b_{-i}^T) \right] = u_i'(\hat{v}_i, \hat{v}_{-i}) + o(1) \,,$$

thus, since $A'$ is truthful, this quantity is maximized for $\hat{v}_i = v_i$, up to the $o(1)$ term. $\qquad\square$

# F   Omitted Details from Section 5

*Proof of Proposition 5.1.* Let $A_T = p_T \cdot A + (1 - p_T) \cdot \text{SPA}$, where $A$ is some auction with $\gamma_A > 0$ and some $p_T$ that will be defined shortly. Notice that

$$\gamma_{A_T} \geq p_T \cdot \gamma_A + (1 - p_T) \cdot \gamma_{\text{SPA}} \geq p_T \cdot \gamma_A \,.$$

Since the bidders are mean-based no-regret learners, we know that when

$$\sum_{\tau=1}^{T_0} v_i \cdot x_i(v_i, b_\tau) - p_i(v_i, b_\tau) \geq \sum_{\tau=1}^{T_0} v_i \cdot x_i(b', b_\tau) - p_i(b', b_\tau) + \delta_T \cdot T, \forall i \in \{0, 1\}, \forall b' \in B_\Delta \,,$$

they will be bidding truthfully with probability at least $1 - \Delta \cdot \eta_T$. We know that in every round

$$v_i \cdot x_i(v_i, b_\tau) - p_i(v_i, b_\tau) \geq v_i \cdot x_i(b', b_\tau) - p_i(b', b_\tau) + \gamma_{A_T}$$
$$\geq v_i \cdot x_i(b', b_\tau) - p_i(b', b_\tau) + p_T \cdot \gamma_A, \forall i \in \{0,1\}, b_\tau, b' \in B_\Delta^2, b' \neq v_i$$

Thus, we define $T_0 = \min\{t \in \mathbb{N} : p_T \cdot \gamma_A \cdot t \geq \delta_T \cdot T\} = \delta_T \cdot T / p_T \cdot \gamma_A$. The regret is

$$\widetilde{\mathrm{Reg}}_T(A_T; v_L, v_H) = \widetilde{\mathrm{Reg}}_{T_0}(A_T; v_L, v_H) + \left( \sum_{t=1}^{T} \mathrm{Rev}(v_L, v_H; \mathrm{SP}) - \mathbf{E}\left[ \sum_{t=T_0+1}^{T} \mathrm{Rev}(b_L^t, b_H^t; A) \right] \right)$$

$$\leq v_L \cdot T_0 + v_L \cdot (T - T_0) \cdot (2\Delta \cdot \delta_T) \cdot (1 - p_T) + (T - T_0) \cdot p_T \cdot v_L$$

$$\leq v_L \cdot (T_0 + 2\Delta \cdot \delta_T \cdot T \cdot (1 - p_T) + T \cdot p_T)$$

$$\leq v_L \cdot \left( \frac{\delta_T \cdot T}{p_T \cdot \gamma_A} + 2\Delta \cdot \delta_T \cdot T + p_T \cdot T \right)$$

$$\leq v_L \cdot \left( \frac{2\Delta \cdot \delta_T \cdot T}{p_T \cdot \gamma_A} + p_T \cdot T \right),$$

where the first inequality follows from the fact that after the first $T_0$ rounds the auctioneer regret is bounded the sum of the probabilities that the auction is SPA and the bidders do not bid truthfully, which is at most $(1 - p) \cdot 2\Delta \cdot \eta_T$, and the probability that auction is not SPA, which is $p_T$. The rest of the inequalities are just algebraic manipulations. Thus, by setting $p_T = \sqrt{2\Delta \cdot \delta_T / \gamma_A}$ we get that

$$\widetilde{\mathrm{Reg}}_T(A_T; v_L, v_H) \leq v_L \cdot \left( 3 \cdot \sqrt{\frac{2\Delta \cdot \delta_T}{\gamma_A}} \cdot T \right),$$

which concludes the proof. $\qquad\square$

*Proof of Proposition 5.2.* Consider the $v_L, v_H$ pairs of the form $v_H = v_L + 1/\Delta$, such that both are bounded away from 0 and 1. Then, Myerson's payment formula shows that $p_H(v_H, v_L) \leq (v_H - 1/\Delta) \cdot x_H(v_H, v_L) = v_L \cdot x_H(v_H, v_L)$. We first argue that $x_H(v_H, v_L) < 1$. Indeed, suppose that $x_H(v_H, v_L) = 1$. Then the low type gets no signal about their bid and hence bids uniformly at random between $[0, v_L]$. In particular, with some $C_\Delta$ probability that is independent of $T$, the low type bids the value $b_L = v_L/2$. Now the only way for the auction $A_T$ to generate $(v_L - o(1))$ revenue from such rounds is if $x_H(v_H, v_L/2) - x_H(v_L, v_L/2) = 1 - o(1)$. But if this is the case, then consider the valuation pair $(v_L/2, v_L/2 + 1/\Delta)$: the auctioneer allocates at most $x_H(v_L/2 + 1/\Delta, v_L/2) \leq o(1)$ per round, and gets almost no revenue from the high type. Moreover, the low type will generate at most $v_L/2$ revenue, so the the regret of the auctioneer is at linear in $T$; this gives the desired contradiction.

Since $x_H(v_H, v_L) < 1$, let $q := 1 - x_H(v_H, v_L)$. Then, $x_L(v_L, v_H) \leq q$ and so $u_L(v_L, v_H) - u_L(v_L - 1/\Delta, v_H) \leq q \cdot 1/\Delta \leq q$. In order to cancel the effect of the learning rate of $\eta_T$, we need to wait for $T_0 := \Omega(1)/(q \cdot \eta_T)$ rounds. For some $C'_\Delta$ fraction of these $T_0$ rounds the agent of low type will bid $v_L/2$, and an argument similar the previous paragraph shows that the revenue of the auction will be at least $1/\Delta - o(1)$ less than $v_L$. Thus, the regret in these $T_0$ rounds will be $\Omega(T_0)$, where we are hiding constants depending on $\Delta$. Let us assume that after $T_0$ rounds the low type starts bidding truthfully. Then, the total regret in this period due to allocation of the item to the low type is $\Omega((T - T_0) \cdot q)$. Summing up the two terms we get a regret of $\Omega(1/(q\eta_T) + q \cdot T - 1/\eta_T)$. Since $\eta_T = \Theta(1/\sqrt{T})$, this is $\Omega(\sqrt{T}/q + qT - \sqrt{T})$, which for any choice of $q$ is $\Omega(T^{3/4})$. $\qquad\square$

*Proof of Theorem 5.3.* We will upper bound the auctioneer regret in the two epochs $\{1, \ldots, T_0\}$, and $\{T_0 + 1, \ldots, T\}$, separately, where $T_0 \in [T]$ is a parameter of the design which we will define shortly. For the first epoch, we will use the simple upper bound of $v_L \cdot T_0$.

Let us consider the bid distribution of the two bidders after $T_0$ rounds. Since they are mean-based no-regret learners we know that if

$$\sum_{\tau=1}^{T_0} v_i \cdot x_i(v_i, b_\tau) - p_i(v_i, b_\tau) \geq \sum_{\tau=1}^{T_0} v_i \cdot x_i(b', b_\tau) - p_i(b', b_\tau) + \delta_T \cdot T, \forall i \in \{1, 2\}, \forall b' \in B_\Delta,$$

then, by a union bound over the possible bids, they will both be bidding truthfully with probability at least $1 - 2\Delta \cdot \eta_T$.

We know that in every round $\tau \in [T_0]$ we have that

$$v_i \cdot x_i(v_i, b_\tau) - p_i(v_i, b_\tau) \geq v_i \cdot x_i(b', b_\tau) - p_i(b', b_\tau) + \gamma_A, \forall i \in \{0, 1\}, b_\tau, b' \in B_\Delta^2, b' \neq v_i.$$

Therefore, we set $T_0 = \min\{t \in \mathbb{N} : t \cdot \gamma_A \cdot t \geq \delta_T \cdot T\} = \delta_T \cdot T / \gamma_A$. Thus, we can upper bound the cumulative auctioneer regret by

$$\widetilde{\text{Reg}}(A, \ldots, A, \text{SPA}, \ldots, \text{SPA}; v_L, v_H) \leq v_L \cdot T_0 + v_L \cdot (T - T_0) \cdot 2\Delta \cdot \eta_T$$

$$\leq v_L \cdot \frac{\delta_T \cdot T}{\gamma_A} + v_L \cdot T \cdot 2\Delta \cdot \eta_T$$

$$= O\left(\delta_T \cdot T \cdot \left(\frac{1}{\gamma_A} + \Delta\right)\right),$$

where the first inequality follows from the fact that with probability at most $2\Delta \cdot \eta_T$ one of the two bidders will not be truthful in the last $(T - T_0)$ rounds, and the other inequalities are just algebraic manipulations. $\qquad\square$

*Proof of Proposition 5.4.* It is not hard to see that in the setting we are working on the auctioneer cannot have negative auctioneer regret in any interval of the interaction. For instance, when $v_H = v_L - 1/\Delta$, the SPA performs optimally. Since every $A_t, t \in [T]$, is a truthful auction, Myerson's lemma shows that

$$u_i^t(v_i, b_{-i}) - u_i^t(b', b_{-i}) = \int_{z=b'}^{v_i} x_i^t(z, b_{-i})dz - (v_i - b') \cdot x_i^t(b', b_{-i}), \forall i \in \{1, 2\}, \forall v_i, b', b_{-i} \in B_\Delta^3,$$

so for $b' = v_i - 1/\Delta$ we get that

$$u_i^t(v_i, b_{-i}) - u_i^t(v_i - 1/\Delta, b_{-i}) \leq \frac{1}{\Delta}, \forall v_i, b', b_{-i} \in B_\Delta^3.$$

Thus, in every iteration the utility gain of bidding $v_i$ is at most $1/\Delta$ greater than bidding $v_i - 1/\Delta$. Summing up over the first $T_0$ iterations, we get that

$$\sum_{t=1}^{T_0} \left(u_i^t(v_i, b_{-i}) - u_i^t(v_i - 1/\Delta, b_{-i})\right) \leq \frac{T_0}{\Delta}, \forall v_i, b', b_{-i} \in B_\Delta^3.$$

Let us now shift our attention to the weights that MWU puts on $v_i - 1/\Delta, v_i$, after $T_0$ iterations. We have

$$\frac{\mathbf{Pr}[b_i^{T_0} = v_i]}{\mathbf{Pr}[b^{T_0} = v_i - 1/\Delta]} = e^{\eta_T \sum_{t=1}^{T_0} \left(u_i^t(v_i, b_{-i}^t) - u_i^t(v_i - 1/\Delta, b_{-i}^t)\right)}$$

$$\leq e^{\eta_T \cdot \frac{T_0}{\Delta}},$$

so for $T_0 = \Delta / \eta_T$ we have that

$$\mathbf{Pr}[b^{T_0} = v_i - 1/\Delta] \geq \frac{\mathbf{Pr}[b_i^{T_0} = v_i]}{e}.$$

This immediately implies that

$$\mathbf{Pr}[b^t = v_i - 1/\Delta] \geq \frac{\mathbf{Pr}[b_i^t = v_i]}{e}, \forall t \in [T_0].$$

Thus, the probability of bidding truthfully of both algorithms is bounded by $9/10$. Thus, when $v_H = v_L + 1/\Delta$ when both bidders are not bidding truthfully the revenue loss compared to SPA is at least $1/\Delta$. Putting it together, we can see that within the first $T_0$ rounds the total revenue loss compared to SPA is at least $C \cdot 1/\Delta \cdot T_0 = C \cdot \eta_T = C \cdot \sqrt{T}$, for some absolute constant $C > 0$. $\qquad\square$

Next, we show that the auction we defined in Definition 5.5 is optimal, in terms of its parameter $\gamma_A$.

**Lemma F.1.** *In the setting with two bidders it holds that the optimal choice of the parameter $\gamma_A$ is $\Theta\left(1/\Delta^2\right)$. Moreover, the auction defined in Definition 5.5 achieves that bound.*

*Proof of Lemma F.1.* Consider some auction $A$ and fix the bid of the second bidder to be $b' \in B_\Delta$. Then, $x_1(\cdot, b')$ is a non-decreasing function, with $0 \le x_1(b, b') \le 1, \forall b \in B_\Delta$. Notice that for any consecutive bids, Myerson's lemma shows that

$$u_1(b, b') - u_1(b - 1/\Delta, b') \le 1/\Delta \cdot (x_1(b, b') - x_1(b - 1/\Delta, b')) \,.$$

Since there are $1/\Delta$ different $b \in B_\Delta$ and the function $x_1(\cdot, b')$ is monotone and bounded between $[0, 1]$ we have

$$\sum_{b > 0} x_1(b, b') - x_1(b - 1/\Delta, b') = x_1(1, b') - x_1(0, b')$$

$$\le 1 \,,$$

and since there are $1/\Delta$ terms in the summation, all of which are non-negative at least one of them must be at most $1/\Delta$. Let $b_1^* \in B_\Delta$ be such that $x(b_1^*, b') - x(b^* - 1/\Delta, b') \le \frac{1}{\Delta}$. Then, picking $v_1 = b_1^*$ witnesses that $\gamma_A \le \frac{1}{\Delta^2}$.

$\square$

# G  Extensions

In this section we discuss potential extensions of our model and adaptations of our results.

**Extension to partial feedback setting.**    Our results can be adapted to the partial feedback setting, with different quantitative bounds. In particular, there are mean-based no-regret algorithms such as EXP3 (Braverman et al., 2018) with $\eta_T = \widetilde{O}(T^{1/4})$. Notice that our positive results are stated for mean-based learners, so the guarantees hold in this setting as well.

**Extension to multiple bidders.**    We underline that our results in Section 4 are already stated and proven for multiple bidders. For our upper bounds in Section 5 there is a $1/n$ degradation to the auctioneer regret bound. When we are dealing with $n$ bidders we can create a strictly IC auction $A$ by building upon our "staircase auction" approach for two bidders in the following way: we select some bidder $i \in [n]$ uniformly at random (independently of their bids) and then we allocate to bidder $i$ with probability $b_i$. Thus, for each bidder $i \in [n]$ their allocation probability $x_i(b)$ is a linear function with $x_i(0) = 0, x_i(1) = 1/n$. Hence, Myerson's lemma shows that $u_i(v_i) - u_i(v_i - 1/\Delta) = \Theta(1/(n\Delta^2))$, thus, $\gamma_A = \Theta(1/(n\Delta^2))$. Recall that in the two-bidder case we have shown that this auction gives $\gamma_A = \Theta(1/\Delta^2)$, so the degradation in $\gamma_A$ by $1/n$ leads to a degradation of the same factor in the auctioneer regret compared to the two-bidder setting.

**Extension of regret bounds to the distributional setting.**    In Section 5 we consider a setting where the auctioneer does not have any distributional knowledge about the valuation of the bidders. Notice that our lower bounds are witnessed by valuation pairs of the low type, high type, of the form $v_L = v, v_H = v + 1/\Delta$. Let us now consider a distributional setting where $v_1, v_2$ are drawn from distributions $\mathcal{D}_1, \mathcal{D}_2$, and then the two bidders participate in repeated second-price auctions using MWU parametrized by these valuations. Similarly as in the prior-free setting, the goal of the auctioneer is to have small expected regret, where the expectation is over the random draw of the valuations and the random behavior of MWU. Notice that the cumulative revenue of SPA when the bidders are truthful is $T \cdot \mathbb{E}_{v_1 \sim \mathcal{D}_1, v_2 \sim \mathcal{D}_2}[\min\{v_1, v_2\}]$, so this is the benchmark the auctioneer competes with (in this setting, we can modify the benchmark to be SPA with personalized reserves with the same arguments). If these distributions $\mathcal{D}_1, \mathcal{D}_2$, place some constant probability (i.e., independent of $T$) on every element of $\{0, 1/\Delta, 2/\Delta, \ldots, 1\}$ then with some constant probability we will see a draw of the form $v_L = v, v_H = v + 1/\Delta$, so these pairs will be contributing a constant fraction of the expected revenue of the second-price auction, i.e., the term $\mathbb{E}_{v_1 \sim \mathcal{D}_1, v_2 \sim \mathcal{D}_2}[\min\{v_1, v_2\}]$. Thus, if the auctioneer wants to have expected regret at most $O(R_T)$, they need to have regret at most $O(R_T)$ for all such valuation pairs, where in the notation $O(\cdot)$ we are suppressing all the parameters that do not depend on $T$.

