# OpenReview forum: "Randomized Truthful Auctions with Learning Agents"
_NeurIPS.cc/2024/Conference — NeurIPS 2024 poster_

### Official Review · Reviewer_9Ayy · 2024-07-08

**Soundness:** 4
**Presentation:** 4
**Contribution:** 4
**Rating:** 7
**Confidence:** 4

**Summary:**

The paper considers repeated auctions with agents using no-regret learning algorithms. It first extends previous result on second price auction to all deterministic auctions that  the runner-up bidder may not converge to bidding truthfully, and provides the condition how the learning rates of the bidders affect the convergence of the bidders. Then it is shown that with bidders being learning agent, randomized auctions can have strictly better revenue guarantees than second price auctions with reserves. Finally, a notion of auctioneer regret is defined, measuring the revenue guarentee in learning agent settings comparing to second price auction with truthful bids, and corresponding regret bounds are provided for auctioneers using the same auction and various auctions.

**Strengths:**

1. The results on biding convergence and the revenue of auctions with respects to bidders using learning algorithms are important.
2. The idea of using auctioneer regret to analyze the revenue guarentee in learning agent settings is interesting and inspiring.
3. The paper provides concrete theoretical proofs.

**Weaknesses:**

1. MWU is the only learning algorithm that considered for bidders. Although it is resresentative, it would be better see more general results for other learning algorithm, or more general characterization on learning algorithms.

**Questions:**

NA

---

> ### Author Rebuttal · Authors · 2024-08-05
>
> We would like to thank the Reviewer for their detailed feedback. Please find our answers to your questions below.
>
> > MWU is the only learning algorithm that considered for bidders. Although it is representative, it would be better see more general results for other learning algorithm, or more general characterization on learning algorithms.
>
> Notice that our results in Sections 4, 5 hold for all mean-based algorithms, which generalize MWU. We will emphasize that more in the next version of our work.

---

> > ### Comment · Reviewer_9Ayy · 2024-08-13
> >
> > Thanks for your response.

---

### Official Review · Reviewer_Lf7G · 2024-07-15

**Soundness:** 3
**Presentation:** 3
**Contribution:** 3
**Rating:** 6
**Confidence:** 4

**Summary:**

This work studies a setting where bidders use no-regret learning algorithms (e.g., MWU) to participate in repeated auctions. Bidders' values are assumed to be persistent.

Generalizing [Kolumbus and Nisan 2022a]'s results on second-price auctions with two equal-learning rate MWU bidders, the authors show that:

(1) in *all deterministic truthful auctions*, if the learning rate of the runner-up bidder is asymptotically equal to or slower than the winning bidder, then the bidders will *not* converge to truthful bidding (which hurts the auctioneer's revenue).

(2) for *some* deterministic truthful auctions (not *all*, see my (Q2)), if the learning rate of the runner-up bidder is asymptotically faster than the winning bidder, then the runner-up bidder converges to truthful bidding (while the winning bidder does not).

Then, the authors design auctions to maximize the auctioneer's revenue, aiming to achieve no-regret against the second-price auction revenue. The basic idea is to take the mixture of the revenue-maximizing IC auction and a randomized strictly IC auction to ensure that bidders can converge to truthful bidding. For the finite horizon setting, the authors obtain a tight regret bound of $\Theta(\sqrt{T^{3/4}})$ by using a constant auction throughout $T$ rounds, and a tight regret bound of $\Theta(\sqrt T)$ by using an adaptive auction schedule.

**Strengths:**

(S1) [Significance & Originality] The characterization of bidders' learning outcomes under different learning rates is very interesting. It is a significant generalization of previous work [Kolumbus and Nisan 2022a] that only considers equal learning rates. Result (1) above holds for all deterministic truthful auction, which is also a significant generalization of [Kolumbus and Nisan 2022a] on second-price auction only.

(S2) [Quality] The auctioneer's regret bounds in the finite horizon analysis are tight (in $T$), which is good.

(S3) [Clarity] The writing is clear in general. I like the discussion of high-level ideas and intuitions.

**Weaknesses:**

(W1) Bidders having persistent values is a strong assumption. Under this assumption, the results in Sections 4 and 5 (designing auctions to achieve no regret for the auctioneer) are relatively straightforward. Moreover, the recent paper [Cai et al 2023, Selling to Multiple No-Regret Buyers] has already studied the problem of auction design against multiple no-regret learning buyers with iid valuations across time, which seems to be a more natural and challenging setting than the persistent value setting here. Given [Cai et al 2023], this paper's additional contribution is limited.

(W2) The conclusion that "if the learning rate of the runner-up bidder is strictly faster than the learning rate of the winning bidder, then the runner-up bidder converges to bidding truthfully" seems to only hold for some deterministic truthful auctions, not "all deterministic truthful auctions" as claimed by the authors. See my question (Q2).

I lean towards rejection for now due to the above concerns, but may change opinions based on the authors' response.

**Questions:**

(Q1) What's the difference/improvement of this work compared to [Cai et al 2023]? (See W1)?

(Q2) This is an important question regarding the correctness of a result. In Section 3, the authors claim that for *all* deterministic truthful actions, "if the learning rate of the runner-up bidder is strictly faster than the learning rate of the winning bidder, then the runner-up bidder converges to bidding truthfully (Line 213)". However, the formal result in Theorem D.3 and the proof are presented only for *second price auctions*. In fact, if the auction is a trivial auction that always allocates the item to bidder 1 (which is deterministic and weakly truthful), then all bids are the same for bidder 2 and hence bidder 2 cannot learn to converge to truthful bidding, so the authors' claim does not hold. I think the claim only holds for deterministic auctions where the low-value bidder can win by truthful bidding when the high-value-bidder submits a small enough bid, like the second-price auction.  A formal definition of such auctions is needed and a complete proof should be provided. Can the authors respond to this issue?

(Q3) (minor clarification question) Section 4 assumes that a strictly IC auction $A'$ always exists. Does it always exist? Is the "staircase auction" in Definition 5.5 a strictly IC auction? If yes, then maybe mention it in Section 4.

(Q4) What's the intuition that as the discretization is finer ($\Delta$ is larger), the regret of the auctioneer becomes larger, in the order of $O(\Delta^2 \sqrt T)$ ? (Corollary 5.6) This feels a bit counterintuitive.

**Limitations:**

**Suggestions:**

(1) Typo: Line 119, "in the prior-free"

(2) Typo: Line 237, "to a bidding truthfully"

(3) In Section 4, I'd suggest to elaborate on the black-box transformation from IC auctions to strictly IC auctions, and shorten the paragraph about "Equilibrium of Meta-Game in Repeated Strictly IC Auctions". For example, you can move Theorem E.1 (which is referred to in the following paragraphs) from appendix to here.  You can also clarify the existence of strictly IC auctions here (see my question (Q3)).

(4) Line 372: "number of actions $\Delta$" -> "number of discretized bids $\Delta$"

(5) Typo: Line 388: "theoptimal"

(6) Line 559 in Theorem D.1: What is "NPT"?

(7) Typo: Line 608: "round $t$" -> " round $i$"

(8) When defining the auctioneer's regret, you compete with the revenue of the second-price auction. I think you can actually compete with the high value $v_H$, which is a stronger benchmark than the second-price revenue. And to achieve no-regret against $v_H$, the following non-oblivious adaptive auction schedule might work: use a strictly IC auction for $T_0$ rounds until the bidders converge to truthful bidding, observe the bidders' values from their truthful bids, and then in the remaining rounds switch to the auction that always allocates the item to the higher-value bidder at a price of $v_H - \frac{1}{\Delta}$. Is that correct?

---

> ### Author Rebuttal · Authors · 2024-08-05
>
> We would like to thank the Reviewer for their detailed feedback. Please find our answers to your questions below.
>
> >  Bidders having persistent ...
>
> We view our results and the setting in which we work as orthogonal to the setting of [Cai et al 2023]. Firstly, they do not restrict themselves to truthful auctions, and for their welfare extraction results, the agents are allowed to overbid. Secondly, in their setting, redrawing valuations i.i.d. in every round helps the learning process (this was also observed in [Feng et al. 2021]). Intuitively, consider two agents and SPA: for every valuation $v$ of player 1, there is some positive probability that player 2’s draw is below $v$, hence player 1 will learn that bidding truthfully is strictly better (in expectation over the other random draw), which leads to the desired bidding behavior. In such a system, randomness is already present due to the draws of the valuations, which helps the convergence to the right bidding behavior.
>
> Our setting of persistent valuations and restricting to truthful auctions, rather than complex mechanisms, is motivated by online ad-auctions. In such settings, multiple auctions are run every second, whereas the valuations of the advertisers may not change much for time scales of a day or a week. Thus, there are typically large intervals of size $T$ where the valuations of the participating agents are persistent. We agree, that the valuations will likely change over longer time horizons, but as long as the persistence is significant, we think our results and the value of randomized mechanisms over deterministic mechanisms will hold. It is an intriguing question to understand the behavior in an intermediate setting, where the valuations vary slowly over time. For this setting, ideas from our work (and from Kolumbus and Nisan (2022), who also consider bidders with persistent valuations) along with those from [Cai et al. 2023] where valuations are drawn in every round, are likely to be useful. Intuitively, the frequency at which the valuations are drawn would affect the amount of randomization we need to add to the auction to help the bidders converge to the right bidding behavior.
>
> Our work also differs from [Cai et al. 2023] in having different conceptual goals: we aim to “restore” the single-shot behavior in natural auctions, such as second-price auctions, in the presence of mean-based learning agents by making minimal modifications to the underlying auction rule. On the other hand, [Cai et al. 2023] aim to exploit the mean-based learning behavior to extract more revenue, and their auctions diverge from the truthful ones we consider in our work. Thus, in our setting, it is clear that reporting the valuation truthfully to the bidding algorithm is an (almost) optimal strategy for the agents (i.e., the so-called “meta-game” considered by Kolumbus and Nisan is truthful), whereas it is not clear to us whether reporting the valuations truthfully to the no-regret algorithms is an optimal strategy in the setting of [Cai et al. 2023].
>
> Finally, a conceptual message of our work is that the key to convergence of the low-type bidder is the presence of enough randomness. If the ranking of the bidders is very stable due to the lack of inherent randomness (i.e., due to infrequent redraws of the valuations), we show that injecting external randomness into the auction induces the desired learning behavior, improves the revenue and restores the property that advertisers can truthfully report their valuations to the learning algorithms. Having persistent valuations is one case of the ranking of the bidders remaining stable over time: studying it allows us to showcase our main ideas, but a central message here is the presence/absence of stability in the rankings of the bidders is key. We have tried to make this point in Lines 88-93 of our manuscript; we will emphasize it more in the next revision.
>
> >  The conclusion that ...
>
> Thank you for pointing this out, we apologize for the confusion. Our Remark 2 in the Appendix (lines 674-678 of the submitted file) describes exactly the condition you are referring to. Under this condition, the proof provided in the appendix goes through for all such auctions. Notice also that if the runner-up bidder loses no matter what the opponent bids, then regardless of their learning rate they will converge to bidding uniformly at random. Thus, we can indeed characterize the convergence behavior for all deterministic auctions, and all learning rates. We will change the discussion in the main body, mention that this result holds for non-trivial auction  (for trivial ones we have convergence to uniform bidding), and change Remark 2 into a definition that states this property. Moreover, we will modify the proof to formally capture this setting.
>
> > (minor clarification question)...
>
> Indeed, the “staircase auction” does satisfy this condition. We will spell it out in the next version of our work.
>
> > What's the intuition ....
>
> Intuitively, if there are more bids, the strictly IC auction needs to “hedge” against more pairs of valuations of the agents and the strictly IC parameter decreases (i.e., the benefit of bidding truthfully decreases). Thus, we need to run this auction for a larger period of time to induce the desired behavior. In the setting of online ad auctions the number of bids is significantly smaller than the number of auctions, so our focus was to obtain optimal bounds with respect to $T$.
>
> > Suggestions...
>
> We will fix the typos and clarify the black-box transformation in the next revision.
>
> Regarding suggestion (8), this is indeed correct, assuming that we use some *adaptive* auction schedule. However, we want to refrain from using such strategies and stick to non-oblivious ones, due to incentive issues – if the bidders know that we are trying to infer their valuations they should not be reporting truthfully to the learning algorithms. We wish to avoid that and stick to more practical approaches.

---

> > ### Comment · Reviewer_Lf7G · 2024-08-10
> > **Happy with authors' response and raise rating to 6**
> >
> > My concerns are resolved by the authors' response and I raised rating to 6.
> >
> > Given authors' response to W1, I agree with the authors that this paper has a sufficient additional contribution to the literature.
> >
> > And thank you for clarifying my question Q2.  I completely agree that, in the revision, you should "change the discussion in the main body, mention that this result holds for non-trivial auction (for trivial ones we have convergence to uniform bidding), and change Remark 2 into a definition that states this property. Moreover, we will modify the proof to formally capture this setting."  A formal proof is really needed since the devil is always in the details.
> >
> > Another minor suggestion regarding readability:  Since your Section 3 now only presents all results informally and redirect the reader to Appendix D, you might consider adding formal theorem statements there or pointing to the specific theorems (like Theorem D.2, D.3).

---

### Official Review · Reviewer_di38 · 2024-07-15

**Soundness:** 3
**Presentation:** 3
**Contribution:** 3
**Rating:** 7
**Confidence:** 4

**Summary:**

The paper considers the building auction mechanism for settings where the bids supplied by agents are chosen by automated no-regret algorithms operating on their behalf. It has been shown from prior work that when bidding with asymmetric valuations, no-regret algorithms converge to bids substantially far from their valuations even when the auctioneer utilizes a truthful mechanism such has a second-price auction. This leads to scenarios where the bidder with the lower valuations routinely bids lower than their true values resulting a loss of revenue for the auctioneer.

The paper undertakes a deeper study of this phenomenon and suggests methods to remedy this situation. They start by showing that the learning rates of the respective agents plays a substantial role in determining the types of  behavior observed at convergence. They show that when the learning rate of the agent with lower valuations is larger, they converge to truthful bids while this is no longer the case if they use a smaller rate. The most interesting contribution of the paper is the observation that the type of convergent behavior depends on the rewards observed by the learning agents. For instance, when second-price auctions are used, the agent with lower valuation rarely receives rewards as they never win the bid and hence, do not receive any feedback to update their bids. The paper shows that when the auctions are instead \emph{randomized}, that is the auctioneer randomly chooses between say a second-price auction and another truthful mechanism, this allows for convergence to truthful bids irrespective of the specific implementation of the bidding algorithms. When the mixing mechanism is \emph{strictly} incentive compatible, that is, a player obtains \emph{strictly more} utility from truthful bidding, they show that any mean-based no-regret algorithm converges to truthful bidding. Furthermore, the proofs in the paper are natural and easy to follow.

Overall, the paper considers a natural problem and presents an elegant solution. In the process, it also conceptually identifies the counterintuitive behavior of no-regret algorithms in this setting and demonstrates an approach toward remedying this. Furthermore, the technical material in the paper is well-presented and understandable. The main drawback of the results is their restriction to the setting of mean-based algorithms. It would be nice if the authors could comment on whether such a restriction may be removed.

**Strengths:**

See main review

**Weaknesses:**

See main review

**Questions:**

See main review

**Limitations:**

See main review

---

> ### Author Rebuttal · Authors · 2024-08-05
>
> We would like to thank the Reviewer for their detailed feedback. Please find our answers to your questions below.
>
> > The main drawback of the results is their restriction to the setting of mean-based algorithms. It would be nice if the authors could comment on whether such a restriction may be removed.
>
> Our choice of mean-based learners is motivated partly by prior work on learning in auctions, which, to a large extent, deals with mean-based learners, and partly by the fact that this is a broad class of learners which can be quantitatively reasoned about in one stroke. While we believe that our qualitative results should extend to other algorithms, it seems this would have to be a case-by-case analysis; it is not clear what broad general condition to place on a learning algorithm so that our results hold quantitatively.

---

> > ### Comment · Reviewer_di38 · 2024-08-08
> >
> > Thank you for the response! I will retain my current evaluation.

---

### Official Review · Reviewer_rdXf · 2024-07-17

**Soundness:** 3
**Presentation:** 2
**Contribution:** 3
**Rating:** 6
**Confidence:** 4

**Summary:**

This work builds upon Kolumbus and Nisan (2022a), which studies a setting where agents use no-regret learning algorithms to bid in a repeated auction setting. The authors first focus on a deterministic setting with two bidders that use the Multiplicative Weights Update (MWU) algorithm. In this case, they show that the runner-up bidder may not converge to bidding truthfully, depending on the learning rate it uses compared to the other agent. Next, they show that adding randomness to the auction can lead to the truthful bidding of the runner-up agent and, hence, maximize the revenue of the auctioneer. Finally, the authors study the non-asymptotic case.

**Strengths:**

I believe the question of studying repeated auctions where agents use algorithms is timely and interesting. I also appreciate how the authors attempt to extend the results of Kolumbus and Nisan (2022a) to any deterministic auction and also highlighting the importance of incorporating randomness.

**Weaknesses:**

I have a number of questions and concerns regarding the generality of the results and the presentation of the paper.

First, as far as I understand, the results in Section 3 are limited to two bidders that use the MWU algorithm. If I am not mistaken, the results of Kolumbus and Nisan (2022a) are not limited to two bidders and also study other algorithms such as "follow the perturbed leader." I wonder how the authors justify or perceive the limitations of their work in this regard.

Next, the model described in Section 2 (and the results of Section 3) are for two bidders, but it seems that from Section 4 onward, the authors switch to an $n$ bidder case. Is this correct?

**Questions:**

Please seem my comments above. Also, can the authors comment on the persistent value assumption? What if the values are drawn independently in each round but from different fixed distributions (in other words, we have persistent values with some noise)?

**Limitations:**

The limitations are discussed as the modeling assumptions are stated clearly.

---

> ### Author Rebuttal · Authors · 2024-08-05
>
> We would like to thank the Reviewer for their detailed feedback. Please find our answers to your questions below.
>
> > First, as far as I understand, the results in Section 3 are limited to two bidders that use the MWU algorithm. If I am not mistaken, the results of Kolumbus and Nisan (2022a) are not limited to two bidders and also study other algorithms such as "follow the perturbed leader." I wonder how the authors justify or perceive the limitations of their work in this regard.
>
> Regarding the number of bidders in Section 3, we focus on the case of two bidders to keep the presentation cleaner; the results can go through when we have a larger number of bidders, where we let W be the winning bidder if they were all to bid truthfully and R the runner-up bidder if they were all to bid truthfully (i.e., the one who would win in the absence of W). Then, the convergence results about the ratio of the learning rates would apply to these two bidders.
> Notice that the theoretical results of Kolumbus and Nisan hold for the case of two bidders only.
>
> Regarding the choice of the algorithm, notice that the theoretical non-convergence result of Kolumbus and Nisan (Theorem 1 in their paper) applies to bidders who are using MWU. There are some simulations about FTRL, but, to the best of our knowledge, there is no theoretical understanding of the convergence behavior.
> Notice that our transformation in Section 4 considers $n$ bidders who are using mean-based no-regret learning algorithms (a natural class of algorithms that significant prior work has focused on), so these results do apply to Follow-the-Regularized-Leader/Follow-the-Perturbed-Leader type of algorithms.
>
> > Next, the model described in Section 2 (and the results of Section 3) are for two bidders, but it seems that from Section 4 onward, the authors switch to an $n$ bidder case. Is this correct?
>
> This is correct; the results in Section 3, Section 5 consider two bidders, and the transformation in Section 4 applies to $n$ bidders. We are happy to state the model with n bidders, if the reviewer feels that it will make the presentation more transparent.
>
> >  Also, can the authors comment on the persistent value assumption? What if the values are drawn independently in each round but from different fixed distributions (in other words, we have persistent values with some noise)?
>
> The persistent assumption is motivated by online ad-auctions, and is also the main setting that is considered by Kolumbus and Nisan. In the context of online ad auctions, while multiple auctions are run every second, the valuations of the advertisers do not change much for certain time scales, e.g., a day or a week. Thus, there are typically large intervals of size $T$ where the valuations of the participating agents are persistent.  We do agree, however, that the valuations will likely change over longer time horizons. We believe that as long as the persistence is significant, our results and the value of randomized mechanisms over deterministic mechanisms will hold. Having a theoretical analysis showing such behavior for a setting with not fully persistent valuations will be interesting but goes beyond the scope of this work.
> In the model you mentioned, where the valuations are re-drawn in every round but the distributions are different, we believe that the convergence (or non-convergence) would be dictated by the mass that the distributions put on overlapping regions of the supports of the distributions.
>
> At a more conceptual level, we believe that a message of our work is the following: the key to convergence of the low-type bidder is the presence of enough randomness. If the ranking of the bidders is very stable due to the lack of inherent randomness (i.e., due to infrequent redraws of the valuations), our results show that injecting external randomness into the auction induces the desired learning behavior and hence improves the revenue. Moreover, it restores the property that advertisers can truthfully report their valuations to the learning agents that bid on their behalf on the queries. Having persistent valuations is just one case of the ranking of the bidders remaining stable over time: studying this case allows us to showcase our main ideas, but a central message here is the presence/absence of stability in the rankings of the bidders is key. We have tried to make this point in Lines 88-93 of our manuscript; we will emphasize it more in the next revision.

---

> > ### Comment · Reviewer_rdXf · 2024-08-10
> > **Response to the rebuttal**
> >
> > I thank the authors for their detailed response. I am satisfied with the answers provided and keep my score as is.

---

### Decision · Program_Chairs · 2024-09-25

**Decision:**

Accept (poster)

**Comment:**

Reviewers found the problem timely and interesting and the solution elegant. On the negative side, reviewers are generally concerned with (perhaps strong) assumptions and thus the generality of the results. This prevents the paper from being recommended beyond poster acceptance. We hope the authors find the reviews helpful. Thanks for submitting to NeurIPS!